# EFFICIENT MULTI-MODAL DATASET DISTILLATION VIA ANALYTIC PARAMETER MATCHING

## ABSTRACT

Multi-modal dataset distillation (MDD) seeks to compress the large-scale multi-modal data, *e.g.*, images and text, into a compact set of synthetic pairs. Existing methods typically employ a bi-trajectory distillation framework to align the trajectories of expert and student models within each modality. Although effective, this paradigm incurs significant storage and computational overhead due to the large number of checkpoints and the need for double backpropagation, limiting its efficiency and scalability. To overcome these limitations, we propose analytic parameter matching (APM), which directly matches the analytic parameters of the modal projectors rather than the entire trajectory, offering two key advantages: First, instead of storing multiple checkpoints, APM only caches two matrices, which significantly reduces the storage budget. Second, APM avoids the bi-level optimization, as the analytic parameters can be computed in a single forward pass. Theoretically, we establish the connection between these analytic parameters and matrix whitening, clarifying their benefits for MDD. Empirically, APM achieves up to $65\times$ storage reduction, $9.6\times$ distillation speedup, and scales to 1000 synthetic pairs. Extensive experiments on Flickr30k and MS-COCO demonstrate the effectiveness of APM in cross-modal retrieval tasks, *e.g.*, 12.8 IR@1 and 17.8 TR@1 under 100-pairs, outperforming existing MDD methods in most scenarios.

## 1 INTRODUCTION

Dataset distillation (DD) (Wang et al., 2018) has emerged as a *de facto* framework for improving data efficiency and accelerating the training of neural networks (Yu et al., 2024; Lei & Tao, 2024). Traditional DD methods focus on compressing the large-scale vision datasets, *e.g.*, CIFAR (Krizhevsky et al., 2009) and ImageNet-1k (Deng et al., 2009) into smaller yet representative ones. Roughly speaking, these methods can be divided into three categories: Gradient Matching (Zhao et al., 2021; Kim et al., 2022), Trajectory Matching (Cazenavette et al., 2022; Guo et al., 2024), and Statistical Matching (Zhao & Bilen, 2023; Yin et al., 2023; Shao et al., 2024). Recently, the distillation of multi-modal datasets (Wu et al., 2024), *e.g.*, images and text, has drawn increasing attention due to its broader applications in downstream tasks such as cross-modal retrieval and conditional generation.

Existing multi-modal dataset distillation (MDD) methods (Wu et al., 2024; Xu et al., 2024a; Zhang et al., 2025; Dang et al., 2025) adopt trajectory matching (TM) as the distillation framework, where expert trajectories are used to supervise the student models trained on the synthetic dataset. Despite its effectiveness, this framework suffers from two visible drawbacks: First, TM requires storing the entire expert trajectories, *e.g.*, a series of checkpoints $\{\theta_0, \theta_1, \theta_2\}$ in Figure 1, leading to significant storage overhead. For example, LoRS (Xu et al., 2024a) trains 20 trajectories, each containing 10 model checkpoints. This takes up over 30GB of space, even larger than the dataset itself, as shown in Table 1. Second, TM involves double backpropagation during distillation, which first updates the model parameters and then optimizes the synthetic dataset by minimizing the differences between expert and student trajectories, limiting its efficiency and scalability.

Once the weaknesses of existing MDD methods are identified, it is natural to ask: *How can we improve the efficiency and scalability of MDD while preserving its effectiveness?* To answer this question, we first note that the computational bottleneck of MDD stems from the inner model optimization on synthetic datasets. Instead of relying solely on iterative gradient descent, a more efficient alternative is to explore its analytic formulations. In particular, we observe that multi-modal

Figure 1: Schema of TM and APM

Table 1: The storage, time, and space overhead of three MDD methods. APM has a $65\times$ storage reduction and $9.6\times$ speedup over LoRS (Xu et al., 2024a).

| Method | Buffer (Offline) | Distillation (Online) | |
| | Storage (GB) | Time (s/iter) | Space (GB) |
| --- | --- | --- | --- |
| LoRS | 32.6 | 11.50 | 21.78 |
| RepBlend | 14.6 | 1.71 | 10.17 |
| APM | 0.5 | 1.20 | 11.17 |

models with linear modal projectors, *e.g.*, CLIP (Radford et al., 2021), allow us to derive the analytic solutions of their projector parameters by solving a least-squares optimization problem.

Motivated by this insight, we propose analytic parameter matching (APM). For any given model, *e.g.*, $\theta_2$ in Figure 1, APM first computes the analytic parameters of the real and synthetic datasets, and then minimizes their discrepancy to narrow the data distribution gap. Instead of storing the entire trajectory, APM only needs to pre-calculate the analytic parameters of the real dataset, which significantly reduces the storage budget, as shown in Table 1. Furthermore, since these parameters can be obtained directly in the forward pass, APM eliminates the need for double backpropagation, thereby further improving its efficiency and scalability. The contributions of this paper are summarized below:

- We analyze the limitations of existing MDD methods, highlighting their substantial storage overhead due to storing multiple expert trajectories and their inefficiency caused by double backpropagation.

- We propose APM, which replaces the inner model optimization with analytic parameter computation, thereby eliminating trajectory storage and double backpropagation, and improving the efficiency and scalability of MDD.

- Extensive experiments on Flickr30k and MS-COCO demonstrate that APM not only achieves competitive or superior performance compared to state-of-the-art MDD methods, but also reduces storage overhead by up to $65\times$ and time overhead by $9.6\times$ during distillation.

## 2 PRELIMINARIES

Before presenting our method, we introduce some key concepts relevant to this work, including multi-modal contrastive learning and multi-modal dataset distillation. More detailed discussions can be found in Section 5.

**Multi-modal Contrastive Learning (MCL)** aims to learn a shared embedding space across modalities, where semantically matched samples, *e.g.*, an image and its caption, are pulled together, while unmatched samples are pushed apart. Consider an image–text dataset with paired samples $(\boldsymbol{x}_i, \boldsymbol{\kappa}_i) \in \mathcal{D}$, where $\boldsymbol{x}_i$ represents the $i$-th image, and $\boldsymbol{\kappa}_i$ denotes its caption. To project data into the shared space, MCL trains a vision–language model $\mathcal{M} = \{f_\mathrm{E}, f_\mathrm{P}, g_\mathrm{E}, g_\mathrm{P}\}$, where $f_\mathrm{E}$ and $f_\mathrm{P}$ denote the image encoder and projector, and $g_\mathrm{E}$ and $g_\mathrm{P}$ are the text encoder and projector, respectively. Finally, a contrastive learning loss function, *e.g.*, InfoNCE (van den Oord et al., 2018), is adopted to optimize the model. This learning process can be formally described as:

$$\boldsymbol{u}_i = \frac{f_\mathrm{P}\big(f_\mathrm{E}(\boldsymbol{x}_i)\big)}{\big\|f_\mathrm{P}\big(f_\mathrm{E}(\boldsymbol{x}_i)\big)\big\|_2}, \quad \boldsymbol{v}_i = \frac{g_\mathrm{P}\big(g_\mathrm{E}(\boldsymbol{\kappa}_i)\big)}{\big\|g_\mathrm{P}\big(g_\mathrm{E}(\boldsymbol{\kappa}_i)\big)\big\|_2}, \quad \mathcal{L}_\mathrm{NCE} = -\frac{1}{|\mathcal{D}|} \sum_{i=1}^{|\mathcal{D}|} \log \frac{\exp(z_{ii})}{\sum_{j=1}^{|\mathcal{D}|} \exp(z_{ij})}, \quad (1)$$

where $z_{ij} = \boldsymbol{u}_i^\top \boldsymbol{v}_j / \tau$ measures the similarity between image and text, and $\tau$ is a temperature ratio. By narrowing the gap between positive pairs and enlarging the gap between negative pairs, $\mathcal{M}$ can learn the semantic correspondence between images and text, which can be used in downstream retrieval or generation tasks.

**Multi-modal Dataset Distillation** seeks to learn some informative synthetic pairs $\mathcal{S} = \{(\hat{\boldsymbol{x}}_i, \hat{\boldsymbol{\kappa}}_i)\}_{i=1}^{|\mathcal{S}|}$, where $|\mathcal{S}| \ll |\mathcal{D}|$, such that a multi-modal model trained on $\mathcal{D}$ and $\mathcal{S}$ will have comparable perfor-

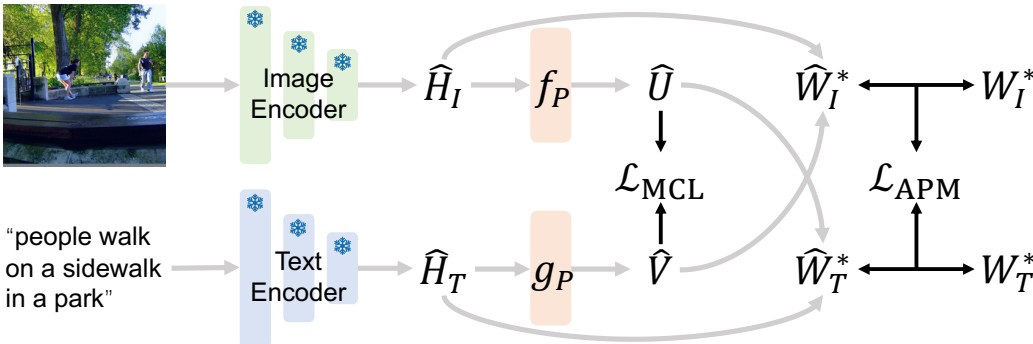

Figure 2: Pipeline of APM. We use the gray arrow to represent the forward pass and the black arrow to denote the calculation of loss functions. $\hat{W}_I^*$ and $\hat{W}_T^*$ are the analytic parameters of the synthetic datasets. The real analytic parameters $W_I^*$ and $W_T^*$ are pre-calculated.

mance. We formulate this task as a bi-level optimization problem:

$$\min_{\mathcal{S}} \sum_{i=1}^{|\mathcal{D}|} \mathcal{L}_{\text{NCE}}(\mathcal{M}^*(\boldsymbol{x}_i, \boldsymbol{\kappa}_i)), \quad \mathcal{M}^* = \arg\min_{\theta} \sum_{i=1}^{|\mathcal{S}|} \mathcal{L}_{\text{NCE}}(\mathcal{M}(\hat{\boldsymbol{x}}_i, \hat{\boldsymbol{\kappa}}_i)), \quad (2)$$

where the inner loop trains the model $\mathcal{M}$ on the synthetic data until convergence, and the outer loop optimizes the synthetic data by minimizing the loss function on the real data. However, solving this bi-level optimization issue is time-consuming. Existing methods (Wu et al., 2024; Xu et al., 2024a) adopt the trajectory matching (Cazenavette et al., 2022) as a surrogate, which minimizes the model optimization trajectories between the real and synthetic data. Despite some efforts, TM requires double backpropagation during training, which greatly limits its efficiency and scalability. This observation motivates the design of our model.

## 3 THE PROPOSED METHOD

In this section, we introduce our proposed method in detail. We begin by deriving the analytic parameters for the image and text projectors, followed by the formulation of the objective function for APM. The overall pipeline of APM is illustrated in Figure 2.

### 3.1 ANALYTIC SOLUTIONS OF MODAL PROJECTORS

To improve the efficiency of MDD, we propose to align the optimal parameters of the modal projectors trained on real and synthetic datasets than their trajectories. The advantages are two-fold: First, we can throw off the massive model checkpoints and focus on distilling the essentials of the dataset. Second, during distillation, we avoid bi-level optimization and only need to propagate the gradient once, which significantly improves the efficiency and scalability of MDD.

However, it is hard to calculate the analytic form of all model parameters due to the non-linearity of neural networks. To solve this issue, we switch to match the image and text projectors as they carry the semantic information across modalities. Specifically, we study the CLIP-style (Radford et al., 2021) network architecture, containing two linear projectors, *i.e.*, $f_P = W_I$ and $g_P = W_T$. For clarity, we use the matrix form to represent the set of $\{\boldsymbol{x}_i\}_{i=1}^{|\mathcal{D}|}$ and $\{\boldsymbol{\kappa}_i\}_{i=1}^{|\mathcal{D}|}$, denoted as $X$ and $K$. As a result, Equation 1 can be reformulated as:

$$H_I = f_{\text{E}}(X) \in \mathbb{R}^{|\mathcal{D}| \times d_I}, \ H_T = g_{\text{E}}(K) \in \mathbb{R}^{|\mathcal{D}| \times d_T}, \ U = H_I W_I, V = H_T W_T \in \mathbb{R}^{|\mathcal{D}| \times d}, \quad (3)$$

where $d_I$ and $d_T$ are the embedding dimensions of image and text, respectively, and $d$ is the dimension of the shared semantic space. Next, we omit the L2-normalization of image and text embeddings, and simplify the MCL loss function as:

$$\mathcal{L}_{\text{MCL}} = \|UV^\top - I\|_F^2 = \|(H_I W_I)(H_T W_T)^\top - I\|_F^2, \quad (4)$$

where $I \in \mathbb{R}^{|\mathcal{D}| \times |\mathcal{D}|}$ is an identity matrix.

**Proposition 1.** *For the linear projectors $U = H_I W_I$ and $V = H_T W_T$, both $\mathcal{L}_{NCE}$ and $\mathcal{L}_{MCL}$ have analytical solutions with respect to $W_I$ and $W_T$, defined as:*

$$W_I^* = \underbrace{\left(H_I^\top H_I\right)^{-1} H_I^\top}_{\textit{Image Whitening}} \underbrace{V \left(V^\top V\right)^{-1}}_{\textit{Text Whitening}}, \quad W_T^* = \underbrace{\left(H_T^\top H_T\right)^{-1} H_T^\top}_{\textit{Text Whitening}} \underbrace{U \left(U^\top U\right)^{-1}}_{\textit{Image Whitening}}. \tag{5}$$

*Proof.* See Appendix B.1. □

**Proposition 2.** *For the non-linear projectors $U = \sigma(H_I W_I)$ and $V = \sigma(H_T W_T)$, the analytic solutions becomes:*

$$W_I^* = (H_I^\top H_I)^{-1} H_I^\top \sigma^{-1}(V(V^\top V)^{-1}), \quad W_T^* = (H_T^\top H_T)^{-1} H_T^\top \sigma^{-1}(U(U^\top U)^{-1}), \tag{6}$$

*where $\sigma^{-1}(\cdot)$ is the inverse function of the activation function $\sigma(\cdot)$.*

*Proof.* See Appendix B.2. □

Since inverse functions may lead to numerical instability, this paper adopts the linear projectors. Building on Proposition 1, the optimal projector for each modality can be decomposed into two important factors: image whitening and text whitening. Here, we take $W_I^*$ as an example, explain the benefit for MDD, and interpret its behind intuition:

- Existing MDD methods focus on the retrieval task, isotropic distributions (Su et al., 2021) are more suitable for cosine similarity-based searches. However, embeddings learned by modal encoders are typically anisotropic. The analytic projectors leverage matrix whitening to address this issue:

$$\left(H_I^\top H_I\right)^{-1} H_I^\top = \underbrace{\left(H_I^\top H_I\right)^{-1/2}}_{\textbf{Whitening matrix}} \underbrace{\left(H_I^\top H_I\right)^{-1/2} H_I^\top}_{\textbf{Whitened embeddings}}, \tag{7}$$

  where the covariance of whitened embeddings is a unit matrix[1], preserving the isotropic property. For text embeddings, we have $\left(V \left(V^\top V\right)^{-1}\right)^\top = \left(V^\top V\right)^{-1} V$, which is also a matrix whitening.

- The goal of DD is to match the distribution between the real and synthetic datasets (Zhao & Bilen, 2023). Hence, we should consider the isotropic distribution as the target, which is easier than the anisotropic one because it does not need to consider the differences between directions. As the optimal parameters of the modal projectors contain whitened embeddings, it is reasonable to use them as a surrogate for the entire model parameters.

## 3.2 ANALYTIC PARAMETER MATCHING

Once the advantages of analytic modal projectors are identified, the next step is to align the distributions of the real and synthetic datasets by matching their analytic parameters. However, there are some instabilities in the calculation of Equation 5: (1) **Embedding Shift.** Matrix whitening requires embeddings to have zero mean (Kessy et al., 2018), but the analytic parameters omit it, which may result in embedding shift. (2) **Scale Explosion.** The whitening matrix involves the sum of sample outer products, *i.e.*, $\left(H_I^\top H_I\right)^{-1/2} = \left(\sum_i \boldsymbol{h}_i^\top \boldsymbol{h}_i\right)^{-1/2}$, which may affect the scale of the analytic parameters of real and synthetic datasets. (3) **Matrix Inversion.** As the size of the synthetic dataset is less than the embedding dimension, the analytic parameters of the synthetic dataset are not full-rank[2]. As a result, directly calculating its inversion may lead to unstable distillation.

---

[1]Here we omit the mean of the embeddings, which will be discussed in Section 3.2

[2]The dimension of embeddings is determined by specific encoder architectures. When NFNet (Brock et al., 2021) and BERT (Devlin et al., 2019) are employed as the image and text encoders, their embedding dimensions are $d_I = 2078$ and $d_T = 768$, respectively, and the maximum size of the synthetic set $|\mathcal{S}|$ is 500.

To overcome these issues, we reformulate the analysis parameters into three parts: image term, cross term, and text term, and introduce some tricks to stabilize the distillation process:

$$\Sigma_{II} = \frac{1}{|\mathcal{D}|}(H_I - \boldsymbol{\mu}_I)^\top(H_I - \boldsymbol{\mu}_I) + \alpha I, \qquad \Sigma_{UU} = \frac{1}{|\mathcal{D}|}(U - \boldsymbol{\mu}_U)^\top(U - \boldsymbol{\mu}_U) + \alpha I, \quad \text{(Image)}$$
$$\Sigma_{IV} = \frac{1}{|\mathcal{D}|}(H_I - \boldsymbol{\mu}_I)^\top(V - \boldsymbol{\mu}_V), \qquad \Sigma_{TU} = \frac{1}{|\mathcal{D}|}(H_T - \boldsymbol{\mu}_T)^\top(U - \boldsymbol{\mu}_U), \qquad \text{(Cross)}$$
$$\Sigma_{TT} = \frac{1}{|\mathcal{D}|}(H_T - \boldsymbol{\mu}_T)^\top(H_T - \boldsymbol{\mu}_T) + \alpha I, \quad \Sigma_{VV} = \frac{1}{|\mathcal{D}|}(V - \boldsymbol{\mu}_V)^\top(V - \boldsymbol{\mu}_V) + \alpha I, \quad \text{(Text)}$$

where $\boldsymbol{\mu}_I$, $\boldsymbol{\mu}_T$, $\boldsymbol{\mu}_U$, and $\boldsymbol{\mu}_V$ denote the mean values of $H_I$, $H_T$, $U$, and $V$, respectively. The hyperparameter $\alpha$ is used to ensure that the matrix is full-rank. Besides, for the synthetic dataset, we use a hat notation to represent their corresponding analytic parameters, $e.g.$, $\hat{\Sigma}_{II}$, and do not repeat their definitions for clarity. The objective function of APM is defined as:

$$\mathcal{L}_{\text{APM}} = \|\Sigma_{II}^{-1}\Sigma_{IV}\Sigma_{VV}^{-1} - \hat{\Sigma}_{II}^{-1}\hat{\Sigma}_{IV}\hat{\Sigma}_{VV}^{-1}\|_F^2 + \|\Sigma_{TT}^{-1}\Sigma_{TU}\Sigma_{UU}^{-1} - \hat{\Sigma}_{TT}^{-1}\hat{\Sigma}_{TU}\hat{\Sigma}_{UU}^{-1}\|_F^2. \qquad (8)$$

It is worth noting that calculating the analytic parameters will introduce quadratic complexity with respect to the number of samples. To avoid this issue, we need to pre-compute the analytic parameters of the real dataset to reduce the time and space overhead during distillation. Therefore, we first pre-train a teacher model $\mathcal{M}^t$ on the real dataset and freeze its weights during the distillation process, so that we can cache the analytic parameters based on the fixed modal encoders and projectors. Finally, we combine the objectives of MCL and APM as the overall loss function for distillation:

$$\mathcal{L} = \sum_{i=1}^{|\mathcal{S}|} \mathcal{L}_{\text{NCE}}(\mathcal{M}^t(\hat{\boldsymbol{x}}_i, \hat{\boldsymbol{\kappa}}_i)) + \eta\mathcal{L}_{\text{APM}}, \qquad (9)$$

where $\eta = 0.01$ is a hyperparameter to balance these two loss functions.

### 3.3 SIMILARITY MINING

In APM, we mainly focus on aligning the channel-level correspondence between the real and synthetic datasets, $i.e.$, the covariance matrices of multi-modal data. On the other hand, mining the correspondence between samples is also crucial for MDD, as pointed out by LoRS (Xu et al., 2024a). Specifically, LoRS uses a LoRA-like (Xu et al., 2024b) matrix, $Z = \omega I + LR^\top$, to record the similarities between samples and optimizes it during distillation.

In the evaluation stage, the similarity matrix is used to weight the binary cross-entropy loss function, aiding the training of multi-modal models. However, this method poses additional computational and space overhead for MDD. Different from LoRS, we directly use the teacher model to generate a similarity matrix of the synthetic pairs rather than training it, and adopt a knowledge distillation loss to train the model from scratch:

$$P_i = \text{Softmax}(\tilde{Z}_i/\tau), \; Q_i = \text{Softmax}(Z_i/\tau), \; \mathcal{L}_{\text{KD}} = \sum_{i=1}^{|\mathcal{S}|} \sum_j P_{ij} \log\frac{P_{ij}}{Q_{ij}}, \qquad (10)$$

where $\tilde{Z} = \mathcal{M}^t(\hat{X}, \hat{K})$ and $Z = \mathcal{M}^s(\hat{X}, \hat{K})$ are the similarity matrices learned by the teacher and student networks. We notice that the similarity matrix of APM is larger than that of LoRS. To address this issue, we can apply SVD on the similarity matrix and preserve eigenvectors corresponding to the top-$K$ singular values.

## 4 EXPERIMENTS

In this section, we conduct extensive experiments to validate the effectiveness of our proposed method. Specifically, we first introduce the experimental setup and then exhibit the quantitative results in Section 4.2. Moreover, we make the ablation studies (Section 4.3) and in-depth analysis (Section 4.4) to further demonstrate the advantages of APM.

### 4.1 EXPERIMENTAL SETUP

**Datasets and Metrics.** Following previous work (Wu et al., 2024; Xu et al., 2024a), we benchmark various MDD methods in two widely used vision-language datasets: Flickr-30k (Plummer et al.,

Table 2: Results on Flickr-30k dataset. We use NFNet+BERT as the distillation and evaluation networks. Full dataset performance: IR@1=21.3, IR@5=51.0, IR@10=63.6; TR@1=31.1, TR@5=61.7, TR@10=74.3. The best results are highlighted in bold.

| Pairs (Ratio) | Metric | Coreset Selection | | | | Dataset Distillation | | | | |
|---|---|---|---|---|---|---|---|---|---|---|
| | | Rand | Herd | K-Cent | Forget | MTT-VL | TESLA | LoRS | RepBlend | APM |
| 100 (0.3%) | IR@1 | 1.0 | 0.7 | 0.7 | 0.7 | 4.7±0.2 | 0.5±0.2 | 8.3±0.2 | 11.5±0.4 | **12.8±0.4** |
| | IR@5 | 4.0 | 2.8 | 3.1 | 2.4 | 15.7±0.5 | 2.3±0.2 | 24.1±0.2 | 32.0±0.7 | **34.2±0.2** |
| | IR@10 | 6.5 | 5.3 | 6.1 | 5.6 | 24.6±1.0 | 4.7±0.4 | 35.1±0.3 | 44.5±0.6 | **47.1±0.3** |
| | TR@1 | 1.3 | 1.1 | 0.6 | 1.2 | 9.9±0.3 | 5.5±0.5 | 11.8±0.2 | 16.2±0.8 | **17.8±0.5** |
| | TR@5 | 5.9 | 4.7 | 5.0 | 4.2 | 28.3±0.5 | 19.5±0.9 | 35.8±0.6 | 41.7±0.9 | **43.0±1.2** |
| | TR@10 | 10.1 | 7.9 | 7.6 | 9.7 | 39.1±0.7 | 28.9±1.0 | 49.2±0.5 | 55.5±0.4 | **57.2±1.1** |
| 200 (0.7%) | IR@1 | 1.1 | 1.5 | 1.5 | 1.2 | 4.6±0.9 | 0.2±0.1 | 8.6±0.3 | 12.7±0.8 | **14.6±0.1** |
| | IR@5 | 4.8 | 5.5 | 5.4 | 3.1 | 16.0±1.6 | 1.3±0.2 | 25.3±0.3 | 34.7±0.6 | **38.5±0.2** |
| | IR@10 | 9.2 | 9.3 | 9.9 | 8.4 | 25.5±2.6 | 2.5±0.2 | 36.6±0.3 | 47.6±0.5 | **52.0±0.3** |
| | TR@1 | 2.1 | 2.3 | 2.2 | 1.5 | 10.2±0.8 | 2.8±0.5 | 14.5±0.5 | 18.6±0.7 | **18.9±1.2** |
| | TR@5 | 8.7 | 8.4 | 8.2 | 8.4 | 28.7±1.0 | 10.4±1.5 | 38.7±0.5 | 46.0±0.8 | **47.8±1.4** |
| | TR@10 | 13.2 | 14.4 | 13.5 | 10.2 | 41.9±1.9 | 17.4±1.6 | 53.4±0.5 | 60.0±0.6 | **62.2±1.1** |
| 500 (1.7%) | IR@1 | 2.4 | 3.0 | 3.5 | 1.8 | 6.6±0.3 | 1.1±0.2 | 10.0±0.2 | 17.0±0.6 | **17.5±0.3** |
| | IR@5 | 10.5 | 10.0 | 10.4 | 9.0 | 20.2±1.2 | 7.3±0.4 | 28.9±0.7 | 42.5±0.5 | **43.5±0.2** |
| | IR@10 | 17.4 | 17.0 | 17.3 | 15.9 | 30.0±2.1 | 12.6±0.5 | 41.6±0.6 | 55.9±0.6 | **56.8±0.3** |
| | TR@1 | 5.2 | 5.1 | 4.9 | 3.6 | 13.3±0.6 | 5.1±0.2 | 15.5±0.5 | **22.5±0.4** | 21.6±0.4 |
| | TR@5 | 18.3 | 16.4 | 16.4 | 12.3 | 32.8±1.8 | 15.3±0.5 | 39.8±0.6 | **53.2±0.3** | 52.7±0.2 |
| | TR@10 | 25.7 | 24.3 | 23.3 | 19.3 | 46.8±3.0 | 23.8±0.3 | 53.7±0.3 | **66.7±0.3** | 66.4±0.4 |

2015) and MS-COCO (Lin et al., 2014), which have 31k and 123k, respectively, and each image is paired with five human-annotated captions. We focus on the cross-modal retrieval task, which aims to retrieve the top-$K$ semantically relevant samples in the target modality conditioned on a query from the source modality. We use Recall at K (R@K) as the metric and consider two scenarios: image-to-text retrieval (TR@K) and text-to-image retrieval (IR@K).

**Preprocessing.** The derivation of the analytic parameters of modal projectors is based on the one-to-one correspondence between images and text. However, in Flickr-30k and MS-COCO, the ratio of the number of images to captions is 1:5, which makes it impossible to directly use APM. To address this issue, we uniformly divide the captions into five datasets and ensure that each image has a corresponding caption. During distillation, we cyclically select one sub-dataset to participate in the calculation of the real analytic parameter, thereby preventing the overfitting of the synthetic dataset.

**Networks.** We use a CLIP-style (Radford et al., 2021) network architecture as our distillation backbone, consisting of an image encoder, a text encoder, and two linear modal projectors. For the image encoder, we choose NFNet (Brock et al., 2021), RegNet (Xu et al., 2023), ResNet-50 (He et al., 2016), and ViT (Dosovitskiy et al., 2021). For the text encoder, we use BERT (Devlin et al., 2019) and DistilBERT (Sanh et al., 2019). We directly optimize the synthetic images in the pixel space and update the embedding of the synthetic captions instead of the original text, as suggested by Wu et al. (2024). We use the officially pre-trained weights to initialize both the image and text encoders. During distillation and evaluation, both encoders are frozen, and we only focus on the modal projectors, as suggested by Zhang et al. (2025).

**Baselines.** We benchmark APM with various MDD methods to demonstrate its effectiveness. Specifically, we consider two categories of methods: Coreset-based methods, including Random, Herding (Welling, 2009), K-Center (Wolf, 2011), and Forgetting (Toneva et al., 2019), as well as the advanced distillation-based methods, including MTT-VL (Wu et al., 2024), LoRS (Xu et al., 2024a), and RepBlend (Zhang et al., 2025).

**Others.** Similar to LoRS, APM also uses the similarity matrix to aid the training of models. To ensure a fair comparison, we remove one synthetic pair to keep the total budget unchanged, *i.e.*, 100→99, 200→199, and 500→499. Moreover, to remove randomness, we evaluate our methods five times and report the mean and standard deviation. See Appendix D for more details, such as hyperparameters and algorithms.

Table 3: Results on MS-COCO dataset. We use NFNet+BERT as the distillation and evaluation networks. Full dataset performance: IR@1=11.1, IR@5=31.5, IR@10=44.7; TR@1=14.6, TR@5=37.6, TR@10=50.5. The best results are highlighted in bold.

| Pairs (Ratio) | Metric | Coreset Selection | | | | Dataset Distillation | | | | |
|---|---|---|---|---|---|---|---|---|---|---|
| | | Rand | Herd | K-Cent | Forget | MTT-VL | TESLA | LoRS | RepBlend | APM |
| 100 (0.8‰) | IR@1 | 0.3 | 0.5 | 0.4 | 0.3 | 1.3±0.1 | 0.3±0.2 | 1.8±0.1 | 4.1±0.3 | **4.7±0.2** |
| | IR@5 | 1.3 | 1.4 | 1.4 | 1.5 | 5.4±0.3 | 1.0±0.4 | 7.1±0.2 | 13.9±0.8 | **16.2±0.2** |
| | IR@10 | 2.7 | 3.5 | 2.5 | 2.5 | 9.5±0.5 | 1.8±0.5 | 12.2±0.2 | 22.3±0.5 | **25.8±0.3** |
| | TR@1 | 0.8 | 0.8 | 1.4 | 0.7 | 2.5±0.3 | 2.0±0.2 | 3.3±0.2 | 5.2±0.5 | **6.2±0.4** |
| | TR@5 | 3.0 | 2.1 | 3.7 | 2.6 | 10.0±0.5 | 7.7±0.5 | 12.2±0.3 | 17.9±0.9 | **20.0±0.5** |
| | TR@10 | 5.0 | 4.9 | 5.5 | 4.8 | 15.7±0.4 | 13.5±0.3 | 19.6±0.3 | 28.0±0.3 | **31.1±0.5** |
| 200 (1.7‰) | IR@1 | 0.6 | 0.9 | 0.7 | 0.6 | 1.7±0.1 | 0.1±0.1 | 2.4±0.1 | **6.1±0.8** | **6.1±0.2** |
| | IR@5 | 2.3 | 2.4 | 2.1 | 2.8 | 6.5±0.4 | 0.2±0.1 | 9.3±0.2 | 19.3±0.7 | **19.6±0.2** |
| | IR@10 | 4.4 | 4.1 | 5.8 | 4.9 | 12.3±0.8 | 0.5±0.1 | 15.5±0.2 | 29.8±0.5 | **30.4±0.3** |
| | TR@1 | 1.0 | 1.0 | 1.2 | 1.1 | 3.3±0.2 | 0.7±0.2 | 4.3±0.1 | 6.9±0.6 | **7.7±0.5** |
| | TR@5 | 4.0 | 3.6 | 3.8 | 3.5 | 11.9±0.6 | 3.1±0.5 | 14.2±0.3 | 21.8±0.9 | **23.6±0.7** |
| | TR@10 | 7.2 | 7.7 | 7.5 | 7.0 | 19.4±1.2 | 5.3±0.8 | 22.6±0.2 | 32.3±0.7 | **35.3±0.9** |
| 500 (4.4‰) | IR@1 | 1.1 | 1.7 | 1.1 | 0.8 | 2.5±0.5 | 0.8±0.2 | 2.8±0.2 | 6.2±0.1 | **7.1±0.2** |
| | IR@5 | 5.0 | 5.3 | 6.3 | 5.8 | 8.9±0.7 | 3.6±0.6 | 9.9±0.5 | 19.9±0.3 | **21.8±0.3** |
| | IR@10 | 8.7 | 9.9 | 10.5 | 8.2 | 15.8±1.5 | 6.7±0.9 | 16.5±0.7 | 30.6±0.1 | **33.3±0.4** |
| | TR@1 | 1.9 | 1.9 | 2.5 | 2.1 | 5.0±0.4 | 1.7±0.4 | 5.3±0.3 | 7.0±0.2 | **8.0±0.4** |
| | TR@5 | 7.5 | 7.8 | 8.7 | 8.2 | 17.2±1.3 | 5.9±0.8 | 18.3±1.5 | 22.0±0.3 | **24.3±0.3** |
| | TR@10 | 12.5 | 13.7 | 14.3 | 13.0 | 26.0±1.9 | 10.2±1.0 | 27.9±1.4 | 32.9±0.6 | **37.1±0.4** |

Table 4: Ablation study on the loss function of APM under 100 pairs. "Random" means we randomly pick data for evaluation without training.

| Flickr | IR@1 | IR@5 | IR@10 | TR@1 | TR@5 | TR@10 |
|---|---|---|---|---|---|---|
| Random | 3.4 | 11.5 | 18.5 | 4.1 | 12.8 | 21.3 |
| $+\mathcal{L}_{MCL}$ | 6.0 | 19.0 | 28.5 | 8.1 | 25.5 | 38.0 |
| $+\mathcal{L}_{APM}$ | **12.8** | **34.2** | **47.1** | **17.8** | **43.0** | **57.2** |

Table 5: Effect of hyperparameters in Flickr-30k.

| IR@1 $\diagdown \eta$ $\alpha$ | 0.1 | 0.01 | 0.001 |
|---|---|---|---|
| 0.01 | 5.2 | 8.1 | 9.9 |
| 0.05 | 7.3 | **12.8** | 11.5 |
| 0.1 | 10.6 | 11.4 | 11.0 |

## 4.2 QUANTITATIVE RESULTS

Tables 2 and 3 report the distillation performance of various MDD methods on Flickr-30k and MS-COCO datasets, from which we have the following observations: First, APM consistently outperforms existing methods in IR. For example, on Flickr-30k with 100 pairs, APM reaches 12.8 IR@1, surpassing RepBlend and LoRS. On MS-COCO with 100 pairs, APM achieves 4.7 IR@1, while RepBlend remains at 4.1. This advantage is not accidental: APM encourages embeddings to be more isotropic and better aligned across modalities, thereby reducing the semantic gap. As a result, it provides consistent improvements on the text-to-image side. Second, APM has a clear improvement in the more challenging MS-COCO benchmark, where the full dataset performance is 11.1 IR@1 and 14.6 TR@1. When the budget increases from 200 to 500 pairs, RepBlend improves IR@1 only marginally (6.1→6.2), while APM gains +0.9 (6.1→7.1). Similar trends hold at higher recall levels, confirming the effectiveness of APM. Third, the improvements are most pronounced under small budgets: on Flickr-30k with 100 pairs, APM improves TR@1 by +5.2 over RepBlend. At larger budgets, the gap narrows. We hypothesize that this is because all methods are initialized with the real dataset, leaving limited room for further improvement.

## 4.3 ABLATION STUDIES

To further verify the effectiveness of each component in APM, we make a series of ablation studies about loss functions, hyperparameters, and cross-architecture generalization.

**Loss Functions.** We first evaluate the role of loss functions, including $\mathcal{L}_{MCL}$ and $\mathcal{L}_{APM}$. The results are shown in Table 4. It can be observed that the randomly selected data outperforms the coreset-based methods, validating the effectiveness of similarity mining. We further add the contrastive loss function $\mathcal{L}_{MCL}$ to optimize the synthetic data and slightly improve the performance, *e.g.*, 3.4→6.0 in

Table 6: Cross-architecture performance of various MDD methods in the Flickr-30k with 500 pairs. The synthetic dataset is distilled on NFNet+BERT and evaluated by other networks.

| Evaluation Model | Method | IR@1 | IR@5 | IR@10 | TR@1 | TR@5 | TR@10 |
|---|---|---|---|---|---|---|---|
| ResNet + BERT | TESLA-VL | 3.0±0.2 | 10.8±0.5 | 17.0±0.8 | 6.0±0.9 | 18.8±0.7 | 27.7±1.2 |
| | LoRS | 3.3±0.2 | 12.7±0.3 | 20.4±0.2 | 6.8±0.2 | 19.6±1.3 | 31.1±0.3 |
| | RepBlend | 4.2±0.2 | 14.1±0.2 | 23.6±0.6 | 8.4±0.2 | 23.1±0.8 | 35.0±1.3 |
| | APM | **6.9±0.2** | **21.2±0.3** | **31.2±0.4** | **8.7±0.7** | **24.5±0.5** | **35.9±1.2** |
| RegNet + BERT | TESLA-VL | 3.2±0.8 | 11.1±1.8 | 17.5±1.3 | 5.8±0.1 | 18.6±0.6 | 28.1±1.0 |
| | LoRS | 3.5±0.1 | 12.6±0.3 | 21.1±0.4 | 6.8±0.3 | 20.8±0.3 | 30.2±0.3 |
| | RepBlend | 3.9±0.2 | 13.9±0.3 | 24.0±0.6 | **7.9±0.3** | **24.2±0.3** | **36.2±1.1** |
| | APM | **5.4±0.1** | **16.7±0.4** | **25.3±0.5** | 7.9±0.5 | 22.2±0.6 | 32.1±0.6 |

IR@1. Finally, we add both $\mathcal{L}_{\text{MCL}}$ and $\mathcal{L}_{\text{APM}}$ in the distillation process, which significantly improves the IR@1 value from 6.0 to 12.8, demonstrating the superiority of APM.

**Hyperparameters.** We next evaluate the influence of hyperparameters on the performance of APM. Specifically, we focus on two important hyperparameters: $\alpha$ in analytic parameters and $\eta$ in loss functions. We can observe from Table 5 that the best result is obtained with $\eta = 0.01$ and $\alpha = 0.05$. Generally, the hyperparameter $\alpha$ controls the frequency of the covariance matrix (Bo et al., 2025). A smaller value of $\alpha$ introduces more high-frequency noise, while a large value of $\alpha$ makes the images blurred. On the other hand, a larger value of $\eta$ may enforce the synthetic dataset to overfit the real analytic parameters, and a smaller value cannot narrow the distribution gap between the real and synthetic datasets.

**Cross-Architecture Generalization.** Finally, we evaluate the cross-architecture generalization of different MDD methods. Following previous work (Zhang et al., 2025), we use NF-ResNet-50 and NF-RegNet as the image encoders, respectively, and BERT as the text encoder. The results are shown in Table 6, from which we can find that APM exhibits the strongest generalization ability across architectures. First, when evaluated on ResNet+BERT, APM achieves the best performance on all metrics, *e.g.*, 6.9 IR@1 and 8.7 TR@1, surpassing RepBlend by +2.7 and +0.3, respectively. Second, on RegNet+BERT, APM consistently outperforms the baselines, reaching 5.4 IR@1 and 7.9 TR@1, while the second-best method only achieves 3.9 and 6.8. This demonstrates that APM not only learns compact and effective synthetic datasets but also transfers well to unseen architectures. The results validate our claim that APM preserves the essential modality alignment in a way that is independent of specific backbone choices, highlighting its scalability and robustness for real-world deployment.

### 4.4 IN-DEPTH ANALYSIS

**Data Entropy.** The goal of DD is to reduce the redundancy in the real datasets. To verify whether APM can achieve this objective, we analyze the entropy of the image and text embeddings in the synthetic dataset. Specifically, we use the SVD entropy, which is defined as $\mathcal{H} = -\sum_i p_i \log p_i$, where $p_i = \frac{\sigma_i}{\sum_i \sigma_i}$ and $\sigma_i$ denote the $i$-th singular value of the data embeddings. Intuitively, data embeddings with smaller SVD entropy have more redundancy as their information is dominated by a few principal singular values, and vice versa. Based on this property, we draw the trends of loss and SVD entropy of the image and text embeddings in Figure 3. We can observe that as the loss function decreases, the SVD entropy of the data embeddings gradually increases, implying that APM can effectively reduce data redundancy and improve data diversity.

**Scalability.** In addition to the efficacy and efficiency, we also emphasize the scalability of the method. Generally, we expect models trained on synthetic datasets to have comparable performance to those trained on real datasets. However, lossless performance is only possible on relatively large-scale synthetic datasets. For example, in the setting of Flickr-30k with 500 pairs, the results of APM are 17.5 in IR@1 and 21.6 in TR@1, which are still far behind the performance on the full dataset (21.3 in IR@1 and 31.1 in TR@1). To evaluate the scalability of APM, we increase the maximum budget from 500 pairs to 1,000 pairs. The results are listed in Table 7. It can be observed that APM achieves the best results in 4 of 6 metrics, while only slightly outperformed by RepBlend in TR@5 and TR@10, demonstrating its scalability.

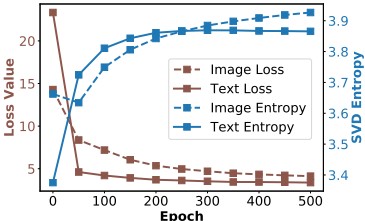

Figure 3: Trends of loss and SVD entropy during distillation.

Table 7: Scalability experiments on Flickr-30k datasets. Results of MDW and EDGE are taken from Dang et al. (2025) and Zhao et al. (2025), while other results were implemented by ourselves.

| Pairs (Ratio) | Method | Flickr-30k | | | | | |
|---|---|---|---|---|---|---|---|
| | | IR@1 | IR@5 | IR@10 | TR@1 | TR@5 | TR@10 |
| 1000 (3.4%) | LoRS | 11.0 | 30.8 | 42.5 | 16.0 | 41.1 | 54.8 |
| | MDW | 12.5 | 32.2 | 45.8 | 19.2 | 49.1 | 63.0 |
| | EDGE | 9.9 | 28.2 | 40.5 | 14.5 | 38.3 | 51.7 |
| | RepBlend | 17.8 | 44.7 | 56.9 | 23.0 | **54.4** | **67.3** |
| | APM | **18.4** | **45.5** | **57.9** | **23.2** | 53.8 | 66.9 |

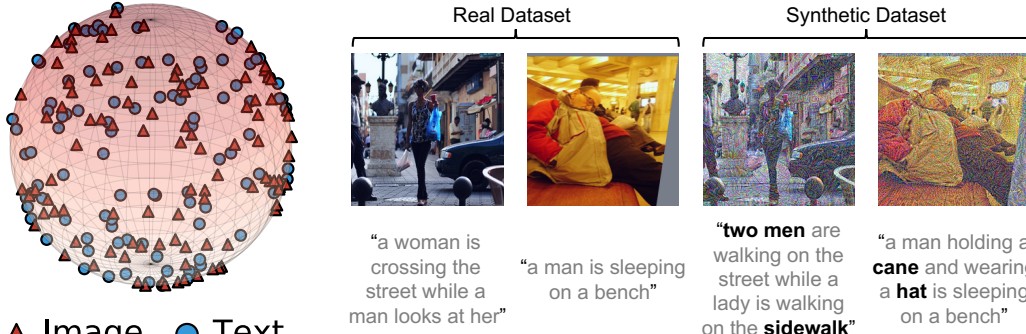

Image   Text

Figure 4: Distribution of the synthetic dataset (Flickr).

Figure 5: The real and synthetic data pairs of APM. We highlight some fine-grained descriptions in the synthetic captions.

### 4.5 VISUALIZATION

**Modality Distribution.** A recent work (Zhang et al., 2025) highlights that MDD methods suffer from the modality collapse issue, where intra-modality embeddings are overly concentrated, while cross-modality embeddings are not well aligned. To verify whether APM can address this issue, we project the image and text embeddings on a spherical surface, as shown in Figure 4. We can observe that the image and text embeddings are well-matched in the shared space, indicating that APM can preserve the data correspondence across modalities.

**Synthetic Pairs.** We compare the real dataset with the synthetic dataset learned by APM in Figure 5. To be more intuitive, the synthetic data pairs are initialized by the real data. It can be observed that the caption learned by APM contains more detailed descriptions, such as the clothing of people. Moreover, the images also have high-frequency artifacts. We speculate that these textures will increase the diversity of data.

## 5 RELATED WORK

**Dataset Distillation.** The concept of dataset distillation (DD) was first introduced by Wang et al. (2018), with the goal of condensing a large dataset into a compact set of synthetic samples while maintaining comparable performance. Existing methods can be broadly categorized into three groups: gradient matching (Zhao et al., 2021; Kim et al., 2022), which aligns gradients computed on real and synthetic data; trajectory matching (Cazenavette et al., 2022; Guo et al., 2024), which supervises the student's optimization trajectory using expert trajectories trained on real data; and statistical matching (Zhao & Bilen, 2023), which aligns higher-order statistics such as feature distributions or batch normalization statistics (Yin et al., 2023; Shao et al., 2024). Moreover, UniDD (Bo et al., 2025) provides a unified spectral filtering view of DD, under which our proposed APM can also be interpreted as a high-pass filter. DD has also been applied across diverse domains, including images (Zhao et al., 2021; Yin et al., 2023), time series (Liu et al., 2024b; Ding et al., 2024), and graphs (Jin et al., 2022; Liu et al., 2024a).

**Multi-modal Dataset Distillation.** Compared with single-modal distillation, the multi-modal setting introduces additional challenges, as it requires preserving both intra-modal semantics and cross-modal alignment. Recent studies have extended trajectory matching (TM) to the multi-modal domain. For instance, MTT-VL (Wu et al., 2024) proposes bi-trajectory matching to align the paired image-text data. LoRS (Xu et al., 2024a) further introduces the concept of similarity mining, improving the performance of MDD by a large margin. More recently, RepBlend (Zhang et al., 2025) identifies the issue of modality collapse in MDD and proposes representation blending to preserve cross-modal consistency. MDW (Dang et al., 2025) further investigates the robustness of MDD under noisy environments. EDGE (Zhao et al., 2025) improves the efficiency and scalability of MDD by leveraging the prior knowledge of generative models.

## 6 CONCLUSION

In this paper, we introduce APM, a framework that improves the efficiency and scalability of multi-modal dataset distillation. APM uses the analytic parameters of linear modal projectors to replace the inner model optimization in trajectory matching, enabling efficient alignment of real and synthetic datasets. Extensive experiments on Flickr30k and MS-COCO demonstrate that APM not only reduces both storage and computational overhead but also maintains superior performance. A promising future direction is to extend APM to other modalities, such as audio-text datasets.

## ETHICS STATEMENT

This work aims to improve the efficiency and scalability of multi-modal dataset distillation, thereby reducing the computational and storage requirements for training large neural networks. There are many potential societal consequences of our work, none of which we feel must be specifically highlighted here.

## REPRODUCIBILITY

To ensure the reproducibility of our work, we introduce more implementation details in Appendix D. During peer reviewing, we upload the source code as supplementary material. We promise to make our code publicly available if the paper is accepted.

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

## A  STATEMENT ON LLM USAGE

In preparing this manuscript, we employed LLMs solely to assist with language polishing and grammar checking. No sections of the text were directly copied from LLM outputs, and all scientific ideas, analyses, and conclusions are original contributions of the authors.

## B  DERIVATION

### B.1  ANALYTIC SOLUTION OF LINEAR PROJECTOR

Recalling that Proposition 1 aims to give the analytic solutions of the linear modal projectors. Here we provide the detailed derivations.

For the loss function $\mathcal{L}_{\mathrm{MCL}} = \|UV^\top - I\|_F^2$, we directly use the gradient of matrix trace to calculate the analytic parameters.

*Proof.* We prove the statement for $W_I$, and the proof for $W_T$ is analogous.

We consider the alternative Optimization of $W_I$ and $W_T$, where we first fix $W_T$ and optimize $W_I$:

$$\mathcal{L}(W_I) = \left\| H_I W_I V^\top - I \right\|_F^2 = \mathrm{tr}\Big( (H_I W_I V^\top - I)^\top (H_I W_I V^\top - I) \Big).$$

Based on the trace of the matrix, we have:

$$\mathcal{L}(W_I) = \mathrm{tr}\big( W_I^\top (H_I^\top H_I) W_I (V^\top V) \big) - 2\,\mathrm{tr}\big( (H_I^\top V)^\top W_I \big) + \mathrm{tr}(I).$$

We now apply standard matrix calculus identities:

$$\frac{\partial}{\partial X}\,\mathrm{tr}(X^\top A X B) = AX(B + B^\top), \quad \frac{\partial}{\partial X}\,\mathrm{tr}(C^\top X) = C.$$

Since both $A = H_I^\top H_I$ and $B = V^\top V$ are symmetric, we obtain:

$$\nabla_{W_I}\mathcal{L} = 2(H_I^\top H_I) W_I (V^\top V) - 2H_I^\top V.$$

Setting the gradient to zero yields the normal equation:

$$(H_I^\top H_I) W_I (V^\top V) = H_I^\top V.$$

Assuming invertibility of $H_I^\top H_I$ and $V^\top V$, the final solution is

$$W_I^\star = (H_I^\top H_I)^{-1} H_I^\top V (V^\top V)^{-1}.$$

□

We further consider the non-linear InfoNCE loss function $\mathcal{L}_{\mathrm{NCE}} = -\frac{1}{|\mathcal{D}|} \sum_{i=1}^{|\mathcal{D}|} \log \frac{\exp(u_i v_i^\top)}{\sum_{j=1}^{|\mathcal{D}|} \exp(u_i v_j^\top)}$, which is more challenging than the linear case, but the conclusion is similar.

We begin by introducing an existing result about the analytic solution of a linear layer with the softmax function in the multi-class classification task:

**Lemma 3.** *The probability that a sample belongs to a certain class is defined as:*

$$p(i|x) = \frac{\exp(xw_i^\top + b_i)}{\sum_{i=1}^k \exp(xw_i^\top + b_i)}, \tag{11}$$

*where $x$ is the sample, and $w_i$ and $b_i$ denote the weight and bias of the $i$-th class, respectively. The analytic solutions of $w_i$ and $b_i$ are defined as:*

$$w_i = \mu_i \Sigma^{-1}, \quad b_i = \ln p_i - \frac{1}{2}\mu_i \Sigma^{-1} \mu_i^\top, \tag{12}$$

*where $p_i$ is the ratio of the $i$-th class, $\mu_i$ is the mean value of the data embedding in the $i$-th class, and $\Sigma = \hat{\Sigma} + \hat{\mu}^\top \hat{\mu} + \sum_i p_i \mu_i^\top \mu_i$.*

*Proof.* See Equations 12-15 in Su (2021). $\qquad\square$

Notably, The InfoNCE loss can also be viewed as a multi-class classification task, where each pair is a class. In this case, $u_i$ can be seen as the sample and $v_i^\top$ denotes the weight in the $i$-th class. Therefore, we can directly obtain their analytic solutions:

$$u_i = \frac{1}{\tau} v_i \Sigma_V^{-1}, \quad v_i = \frac{1}{\tau} u_i \Sigma_U^{-1},\tag{13}$$

where $\Sigma_V$ and $\Sigma_U$ denote the covariance of each modality. Notably, we have $\hat{\Sigma} + \hat{\mu}^\top \hat{\mu} = \frac{1}{N}\sum_i x_i^\top x_i = \frac{1}{N}X^\top X$, $p_i = \frac{1}{N}$, and $\mu_i = u_i$. Therefore, $\Sigma_V = \frac{2}{N}U^\top U$ and $\Sigma_U = \frac{2}{N}V^\top V$.

We then transform this equation into matrix form by stacking a series of vectors, and obtain:

$$U = \frac{N}{2\tau}V(V^\top V)^{-1}, \quad V = \frac{N}{2\tau}U(U^\top U)^{-1}.\tag{14}$$

Based on the above analysis, we can find that both InfoNCE and the least-square loss $\|UV^\top - I\|_F^2$ have similar results, and the softmax function does not affect the calculation of analytic solutions.

### B.2 ANALYTIC SOLUTION OF NON-LINEAR PROJECTOR

For the non-linear projectors $U = \sigma(H_I W_I)$ and $V = \sigma(H_T W_T)$, we have $\sigma(H_I W_I) = V(V^\top V)^{-1}$ and $\sigma(H_T W_T) = U(U^\top U)^{-1}$. Since the activation $\sigma$ is an element-wise function, we can directly use its inversion function to calculate the analytic solutions:

$$W_I^* = (H_I^\top H_I)^{-1}H_I^\top \sigma^{-1}(V(V^\top V)^{-1}), \quad W_T^* = (H_T^\top H_T)^{-1}H_T^\top \sigma^{-1}(U(U^\top U)^{-1}),\tag{15}$$

where $\sigma^{-1}(\cdot)$ is the inverse function of $\sigma(\cdot)$.

We list some commonly used activation functions and their inverses below.

Table 8: Activation Functions $\sigma(x)$ and their inverses $\sigma^{-1}(y)$

| Activation | $\sigma(x)$ | $\sigma^{-1}(y)$ |
|:---:|:---:|:---:|
| Sigmoid | $\sigma(x) = \dfrac{1}{1+e^{-x}}$ | $\sigma^{-1}(y) = \ln\left(\dfrac{y}{1-y}\right)$ |
| Tanh | $\sigma(x) = \tanh(x) = \dfrac{e^x - e^{-x}}{e^x + e^{-x}}$ | $\sigma^{-1}(y) = \operatorname{arctanh}(y) = \dfrac{1}{2}\ln\left(\dfrac{1+y}{1-y}\right)$ |
| LeakyReLU | $\sigma(x) = \begin{cases} x, & x \geq 0 \\ \alpha x, & x < 0 \end{cases}$ | $\sigma^{-1}(y) = \begin{cases} y, & y \geq 0 \\ \frac{y}{\alpha}, & y < 0 \end{cases}$ |

## C ADDITIONAL EXPERIMENTS

### C.1 ISOTROPY AND ANISOTROPY DISTRIBUTIONS

We make an experiment to verify the effectiveness of isotropy distribution.

As mentioned Section 3.1, the analytic parameters $(H_I^\top H_I)^{-1}H_I^\top V(V^\top V)^{-1}$ have a close connection with matrix whitening, which transforms the embeddings into a isotropy distribution.

To construct a anisotropy distribution, we remove the inverse covariance in the analytic parameters and obtain $H_I^\top V$, which is dominated by the principle singular values.

The comparison between the isotropy and anisotropy distributions is shown in Table 9. We have the following observations: First, the synthetic data learned by isotropy distribution has large entropy, indicating that it can encode more information of the real data. Second, the isotropy distribution outperforms anisotropy distribution by a large margin, verifying the effectiveness of isotropy distribution and supporting our claims.

Table 9: Comparison between isotropy and anisotropy distributions.

| Type | Equation | Image Entropy | Text Entropy | IR@1 | TR@1 |
|------|----------|---------------|--------------|------|------|
| Isotropy | $(H_I^\top H_I)^{-1} H_I^\top V (V^\top V)^{-1}$ | 3.93 | 3.87 | 12.8 | 17.8 |
| Anisotropy | $H_I^\top V$ | 3.87 | 3.79 | 7.9 | 11.2 |

## C.2 ZERO-SHOT CLASSIFICATION

To verify whether the synthetic dataset can be used in downstream tasks beyond retrieval. We make a zero-shot image classification task to benchmark the performance between real and synthetic datasets.

Specifically, we use three datasets, CIFAR-10, CIFAR-100, and ImageNet-1k. The results are shown Table 10. We can see that the synthetic datasets have similar zero-shot classification performance with the real dataset.

Table 10: Results of zero-shot image classification.

| Dataset | CIFAR-10 | | CIFAR-100 | | ImageNet-1k | |
|---------|----------|----------|-----------|----------|-------------|----------|
| | Top-1 (%) | Top-5 (%) | Top-1 (%) | Top-5 (%) | Top-1 (%) | Top-5 (%) |
| Full | 58.77 | 92.07 | 16.34 | 38.27 | 7.62 | 19.54 |
| 99 Pairs | 52.44 | 87.54 | 13.53 | 32.19 | 4.28 | 12.50 |
| 199 Pairs | 53.73 | 85.96 | 13.70 | 34.76 | 4.35 | 12.69 |
| 499 Pairs | 55.03 | 90.28 | 14.45 | 36.14 | 5.06 | 14.77 |

## C.3 ABLATION STUDY

In Section 3.2, we mentioned three issues of the analytic parameters, including Embedding Shift (ES), Scale Explosion (SE), and Matrix Inversion (MI), and proposed three corresponding modifications. We further make an ablation study on the Flickr-30k dataset to validate the effectiveness of these modification. The results are shown Table 11. We have the following observations: First, removing ES slightly affects the performance of APM. The reason is that synthetic data is initialized by real data, thus they may have similar mean values. Second, removing SE significantly reduces the retrieval performance as the number of real data is larger than the synthetic data, making the scale of $H_I^\top H_I$ and $\hat{H}_I^\top \hat{H}_I$ different. Third, removing MI cannot obtain the meaningful synthetic dataset. The covariance of the synthetic dataset is low-rank, and directly solving for its inverse matrix will lead to numerical instability.

Table 11: Ablation studies on the design of loss function.

| | IR@1 | IR@5 | IR@10 | TR@1 | TR@5 | TR@10 |
|------|------|------|-------|------|------|-------|
| $\mathcal{L}_{\text{APM}}$ | 12.8 | 34.2 | 47.1 | 17.8 | 43.0 | 57.2 |
| w/o ES | 11.6 | 32.0 | 44.9 | 17.3 | 41.8 | 57.5 |
| w/o SE | 2.1 | 7.5 | 12.9 | 3.9 | 13.3 | 19.9 |
| w/o MI | 0.2 | 0.8 | 1.3 | 0.0 | 0.0 | 0.0 |

## C.4 COMPARISON WITH EDGE

EDGE leverages generative models to address the semantic correlation and diversity issues of existing MDD methods. Notably, we have cited EDGE in the original paper.

EDGE mainly focuses on the large budget setting (500 / 1000 pairs). We report the performance of EDGE and APM under the same settings in the revision. Below is a quick comparison.

Table 12: Flickr Retrieval Results

| Flickr | IR@1 | IR@5 | IR@10 | TR@1 | TR@5 | TR@10 |
|---|---|---|---|---|---|---|
| EDGE-500 | 6.7 | 21.0 | 30.5 | 13.3 | 35.6 | 47.5 |
| APM-500 | **17.5** | **43.5** | **56.8** | **21.6** | **52.7** | **66.4** |
| EDGE-1000 | 9.9 | 28.2 | 40.5 | 14.5 | 38.3 | 51.7 |
| APM-1000 | **18.4** | **45.5** | **57.9** | **23.2** | **53.8** | **66.9** |

Table 13: MS-COCO Retrieval Results

| MS-COCO | IR@1 | IR@5 | IR@10 | TR@1 | TR@5 | TR@10 |
|---|---|---|---|---|---|---|
| EDGE-500 | 1.8 | 6.5 | 11.2 | 2.9 | 9.5 | 15.7 |
| APM-500 | **7.1** | **21.8** | **33.3** | **8.0** | **24.3** | **37.1** |

## C.5   AUDIO-TEXT DATASET DISTILLATION

We make an additional experiment on the audio-text retrieval task to verify the generalization of the proposed method. Following Zhang et al. (2025), we choose the AudioCaps (Kim et al., 2019) dataset, consisting of 49,838 training audios, 495 validation audios, and 975 test audios. We use EfficientAT (mn20_as) (Schmid et al., 2023) as the audio encoder and BERT as the text encoder. Since RepBlend (Zhang et al., 2025) does not introduce its implementation details, we describe our reproduction process below.

**Data Preparation.** The AudioCaps dataset contains files in WAV format. We use the *AugmentMel-STFT* function from EfficientAT to preprocess the audio. We sample the audios in mono at a sampling rate of 32 kHz and then calculate their log-mel-spectrogram in a 25-ms window with a step size of 10 ms. After processing, each audio has a feature map with shape [1, 128, 1000], as shown in Figure 6.

**Distillation.** Instead of distilling the raw audios, we directly synthesize the log-mel-spectrogram to match the input of EfficientAT. The spectrogram has a shape of [1, 128, 1000], which can be seen as an image with channel=1, width=128, and height=1000. Therefore, the code of image-text distillation can be directly transferred to the audio-text datasets. See Table 15 for the hyperparameters.

**Evaluation.** In the test set of AudioCaps, each audio corresponds to 5 captions, which improves its retrieval performance. We train the multi-modal network from scratch and evaluate it on the test set. We repeat the experiments five times and report the average performance and standard deviation.

**Results.** The results are shown in Table 14. We can observe that APM outperforms LoRS and RepBlend, especially in audio retrieval tasks, where it shows a significant performance improvement. Figure 6 illustrates the original and distilled log-mel-spectrograms. We can see that the distilled log-mel-spectrogram has more energy than the original one, indicating that it compresses the knowledge of other audios.

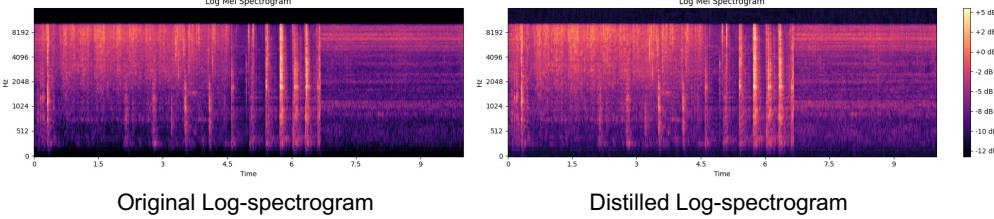

Original Log-spectrogram          Distilled Log-spectrogram

Figure 6: Visualizations of the original and distilled audio data.

Table 14: Results on AudioCaps dataset. We use EfficientAT (mn20_as)+BERT as the distillation and evaluation networks. Full dataset performance: AR@1=17.6, AR@5=47.7, AR@10=63.8; TR@1=20.6, TR@5=49.6, TR@10=67.2. The best results are highlighted in bold.

| Pairs | Method | AR@1 | AR@5 | AR@10 | TR@1 | TR@5 | TR@10 |
|-------|--------|------|------|-------|------|------|-------|
| 100 | LoRS | 2.7±0.3 | 8.6±0.3 | 14.7±0.4 | 5.9±0.3 | 13.0±0.4 | 21.8±0.5 |
| | RepBlend | 4.1±0.2 | 14.2±0.3 | 23.7±0.4 | 8.9±0.1 | 24.3±0.2 | 34.7±0.3 |
| | APM | **8.3±0.3** | **28.6±0.3** | **42.1±0.4** | **11.3±0.6** | **33.4±0.6** | **46.7±0.8** |
| 200 | LoRS | 3.8±0.2 | 14.8±0.2 | 21.8±0.2 | 8.0±0.2 | 21.2±0.2 | 33.1±0.2 |
| | RepBlend | 6.8±0.2 | 20.6±0.2 | 31.4±0.3 | 9.7±0.2 | 29.1±0.4 | 41.2±0.4 |
| | APM | **10.1±0.1** | **32.5±0.3** | **47.3±0.2** | **11.7±0.7** | **35.6±0.8** | **51.1±1.1** |
| 500 | LoRS | 7.1±0.1 | 24.7±0.2 | 36.7±0.2 | 9.2±0.2 | 27.4±0.3 | 41.3±0.3 |
| | RepBlend | 9.7±0.1 | 32.2±0.3 | 46.8±0.2 | **13.8±0.3** | 38.6±0.3 | 54.1±0.4 |
| | APM | **11.4±0.1** | **35.8±0.4** | **51.3±0.3** | 13.6±0.7 | **39.3±0.7** | **54.8±0.5** |

## D EXPERIMENTAL DETAILS

**Hyperparameters.** To improve the reproducibility of our work, we provide the hyperparameters used in both distillation and evaluation stages in Tables 15 and 16.

Table 15: Hyperparameters used in the distillation stage.

| Dataset | Flickr | | | COCO | | | AudioCaps | | |
|---------|--------|--------|--------|--------|--------|--------|-----------|--------|--------|
| Pairs | 100 | 200 | 500 | 100 | 200 | 500 | 100 | 200 | 500 |
| Epoch | 400 | 400 | 400 | 400 | 400 | 400 | 400 | 400 | 400 |
| Optimizer | Adam | Adam | Adam | Adam | Adam | Adam | Adam | Adam | Adam |
| LR | 0.1 | 0.1 | 0.1 | 0.1 | 0.1 | 0.1 | 0.01 | 0.01 | 0.01 |
| Betas | (0.6, 0.9) | (0.6, 0.9) | (0.6, 0.9) | (0.6, 0.9) | (0.6, 0.9) | (0.6, 0.9) | (0.6, 0.9) | (0.6, 0.9) | (0.6, 0.9) |
| $\alpha$ | 0.05 | 0.05 | 0.05 | 0.05 | 0.05 | 0.05 | 0.1 | 0.1 | 0.1 |
| $\eta$ | 0.01 | 0.01 | 0.01 | 0.01 | 0.01 | 0.01 | 0.01 | 0.01 | 0.01 |
| Projector Dim. | 256 | 256 | 256 | 256 | 256 | 256 | 256 | 256 | 256 |

Table 16: Hyperparameters used in the evaluation stage.

| Dataset | Flickr | | | COCO | | | AudioCaps | | |
|---------|--------|--------|--------|--------|--------|--------|-----------|--------|--------|
| Pairs | 100 | 200 | 500 | 100 | 200 | 500 | 100 | 200 | 500 |
| Epoch | 100 | 100 | 100 | 100 | 100 | 100 | 100 | 100 | 100 |
| Optimizer | SGD | SGD | SGD | SGD | SGD | SGD | SGD | SGD | SGD |
| LR | 0.1 | 0.1 | 0.1 | 0.1 | 0.1 | 0.1 | 0.1 | 0.1 | 0.1 |
| Momentum | 0.9 | 0.9 | 0.9 | 0.9 | 0.9 | 0.9 | 0.9 | 0.9 | 0.9 |
| Weight Decay | 0.0005 | 0.0005 | 0.0005 | 0.0005 | 0.0005 | 0.0005 | 0.0005 | 0.0005 | 0.0005 |
| Scheduler | StepLR | StepLR | StepLR | StepLR | StepLR | StepLR | StepLR | StepLR | StepLR |
| Projector Dim. | 256 | 256 | 512 | 256 | 256 | 512 | 256 | 256 | 256 |
| KD Temperature ($\tau$) | 5 | 5 | 10 | 5 | 5 | 10 | 5 | 5 | 5 |

**Algorithms** Algorithm 1 illustrates the distillation process of APM. Algorithm 2 shows the Pytorch-style core code of APM.

**Algorithm 1** Analytic Parameter Matching (APM)

**Input:** Distillation network $\mathcal{M} = \{f_\text{E}, f_\text{P}, g_\text{E}, g_\text{P}\}$, real dataset $\mathcal{D} = (X, K)$, number of iteration $\mathcal{I}$.
**Output:** Synthetic dataset $\mathcal{S} = (\hat{X}, \hat{K})$
1: Feed $\mathcal{D}$ into $\mathcal{M}$, where $H_I = f_\text{E}(X), H_T = g_\text{E}(K), U = g_\text{E}(H_I), V = g_\text{P}(H_T)$
2: Calculate $\Sigma_{II}, \Sigma_{UU}, \Sigma_{IV}, \Sigma_{TU}, \Sigma_{TT}$, and $\Sigma_{VV}$
3: Calculate $W_I^* = \Sigma_{II}^{-1}\Sigma_{IV}\Sigma_{VV}^{-1}$ and $W_T^* = \Sigma_{TT}^{-1}\Sigma_{TU}\Sigma_{UU}^{-1}$
4: **for** iteration $i = 1, \cdots, \mathcal{I}$ **do**
5:      Feed $\mathcal{S}$ into $\mathcal{M}$, where $\hat{H}_I = f_\text{E}(\hat{X}), \hat{H}_T = g_\text{E}(\hat{K}), \hat{U} = g_\text{E}(\hat{H}_I), \hat{V} = g_\text{P}(\hat{H}_T)$
6:      Calculate $\hat{\Sigma}_{II}, \hat{\Sigma}_{UU}, \hat{\Sigma}_{IV}, \hat{\Sigma}_{TU}, \hat{\Sigma}_{TT}$, and $\hat{\Sigma}_{VV}$
7:      Calculate $\hat{W}_I^* = \hat{\Sigma}_{II}^{-1}\hat{\Sigma}_{IV}\hat{\Sigma}_{VV}^{-1}$ and $\hat{W}_T^* = \hat{\Sigma}_{TT}^{-1}\hat{\Sigma}_{TU}\hat{\Sigma}_{UU}^{-1}$
8:      Minimize the discrepancy between analytic parameters based on Equation 8
9: **end for**

**Algorithm 2** PyTorch code of APM

```python
def Conv(img_embed, txt_embed, img_proj, txt_proj, alpha=0.1):
    device = img_embed.device
    N = img_embed.shape[0]

    h_I = img_embed - img_embed.mean(0, keepdim=True)
    h_T = txt_embed - txt_embed.mean(0, keepdim=True)
    h_U = img_proj - img_proj.mean(0, keepdim=True)
    h_V = txt_proj - txt_proj.mean(0, keepdim=True)

    sigma_II = (h_I.T @ h_I) / N + alpha * torch.eye(h_I.shape[1], device=device)
    sigma_IV = (h_I.T @ h_V) / N
    sigma_VV = (h_V.T @ h_V) / N + alpha * torch.eye(h_V.shape[1], device=device)

    tmp = torch.linalg.solve(sigma_II, sigma_IV)
    w_I = torch.linalg.solve(sigma_VV, tmp, left=False)

    sigma_TT = (h_T.T @ h_T) / N + alpha * torch.eye(h_T.shape[1], device=device)
    sigma_TU = (h_T.T @ h_U) / N
    sigma_UU = (h_U.T @ h_U) / N + alpha * torch.eye(h_U.shape[1], device=device)

    tmp2 = torch.linalg.solve(sigma_TT, sigma_TU)
    w_T = torch.linalg.solve(sigma_UU, tmp2, left=False)

    return w_I, w_T
```

