# OpenReview forum: "Efficient Multi-modal Dataset Distillation via Analytic Parameter Matching"
_ICLR.cc/2026/Conference — Submitted to ICLR 2026_

### Official Review · Reviewer_kpau · 2025-10-21

**Soundness:** 3
**Presentation:** 3
**Contribution:** 2
**Rating:** 4
**Confidence:** 3

**Summary:**

This paper introduces Analytic Parameter Matching (APM), a new and efficient method for multi-modal dataset distillation (MDD). MDD aims to compress large datasets of paired data into a much smaller, synthetic set that can train models effectively.

**Strengths:**

The strengths of this paper are:
1. It proposes APM, which replaces the expensive inner-loop model optimization and trajectory storage of previous methods with a direct analytic computation.
2. Compared to previous methods like LoRS, APM achieves up to a 65x reduction in storage and a 9.6x speedup in distillation time by avoiding the need to store checkpoints and perform double backpropagation.
3. Extensive experiments on the Flickr30k and MS-COCO datasets show that APM achieves state-of-the-art or competitive performance in cross-modal retrieval tasks, especially with small synthetic dataset sizes.

**Weaknesses:**

The weaknesses of this paper are:
1. The derivation in Equation 4 simplifies the standard InfoNCE contrastive loss to a least-squares problem, i.e., $\mathcal{L}_{MCL}$. This simplification ignores the temperature parameter $\tau$ and the softmax normalization over negative samples, which are fundamental to modern contrastive learning. The paper does not provide sufficient justification for why this simplified objective is a valid proxy. The performance of InfoNCE is known to be highly sensitive to the number of negative samples and the temperature setting.
2. The entire method is predicated on the image and text projectors ($f_P$, $g_P$) being simple linear transformations. While this holds for the original CLIP architecture, it limits the method's applicability to more complex or future multi-modal models that might employ non-linear projectors (e.g., MLPs) to increase expressive power.
3. To handle datasets like MS-COCO where one image has five captions, the paper creates five sub-datasets and cyclically selects one during distillation to calculate the real analytic parameters. This strategy is presented without justification or comparison to alternatives.
4. The paper identifies both $(H_{I}^{\top}H_{I})^{-1}H_{I}^{\top}$ and $V(V^{\top}V)^{-1}$ as whitening operations. However, the standard definition of a whitening transform (e.g., ZCA whitening) produces data with an identity covariance matrix and typically involves the inverse square root of the covariance matrix (i.e., $(H_{I}^{\top}H_{I})^{-1/2}$). The term $(H_{I}^{\top}H_{I})^{-1}H_{I}^{\top}$ is the Moore-Penrose pseudoinverse of $H_I$. While it does decorrelate the data, its properties are not identical to a full whitening transformation.

**Questions:**

Rebuttal questions:
1. The paper's core theoretical leap is replacing the standard InfoNCE contrastive loss with a least-squares objective i.e., $\mathcal{L}_{MCL}$. What is the theoretical or empirical justification for this simplification? Why should this least-squares objective be considered a valid proxy for the InfoNCE loss, whose performance is known to be highly sensitive to $\tau$ and the number of negative samples?
2. In Section 3.1, the authors derive a clean analytic solution (Eq. 5) . However, in Section 3.2, the authors introduce a significantly more complex objective (Eq. 7) based on matching centered covariance matrices, citing instabilities like embedding shift, scale explosion, and rank deficiency. Were these instabilities empirically observed? Could you quantify the performance degradation when matching the simpler Eq. 5 directly? The jump in complexity from Eq. 5 to Eq. 7 feels substantial and requires strong justification.
3. For datasets like Flickr-30k and MS-COCO, the authors handle the 1:5 image-to-caption ratio by creating five sub-datasets and cyclically selecting one sub-dataset to compute the real analytic parameters. What was the motivation for this specific cyclic strategy?

---

> ### Author Response · Authors · 2025-11-21
> **Response to Reviewer kpau**
>
> We sincerely appreciate your thoughtful feedback and insightful questions.
>
> &nbsp;
>
> > **W1 & Q1: Why the least-square loss is a valid proxy of InfoNCE**
>
> See Conclusion 1 in the general response.
>
> &nbsp;
>
> > **W2: From linear projector to non-linear projector**
>
> See Conclusion 2 in the general response.
>
> &nbsp;
>
> > **W3 & Q3: Data pre-processing of image and text pairs**
>
> A3: In the Flickr-30k and MS-COCO datasets, the image-to-caption ratio is 1:5. We uniformly divide the captions into five datasets and obtain five sub-datasets, where each of them has $N$ image-caption pairs.
>
> The motivations are two-fold:
>
> - **Reducing the complexity of calculating analytic parameters.** The analytic parameters involve the Moore–Penrose pseudoinverse of the image and text embeddings, whose complexity is $\mathcal{O}(N^2d)$. If we use the full data to calculate the analytic parameters, the complexity increases to $\mathcal{O}(k^2N^2d)$, where **$k$ is the image-to-caption ratio**. On the other hand, if we use the sub-dataset, the complexity reduces to $\mathcal{O}(kN^2d)$.
>
> - **Preventing the synthetic dataset from over-fitting.** The essence of analytic parameters matching is aligning some statistical information between the real and synthetic datasets. If the entire dataset is used to calculate the analytic parameters, the synthetic dataset may overfit its statistics, while using the sub-datasets cyclically can avoid this issue. Notably, this strategy has been widely used in the distillation of image datasets [1].
>
> [1] Generalized Large-Scale Data Condensation via Various Backbone and Statistical Matching. CVPR 2024.
>
> &nbsp;
>
> > **W4: Matrix Whitening and Moore-Penrose pseudoinverse**
>
> A4: Thanks for pointing out this insight. Let $H = USV^{\top}$ denote the SVD decomposition of the embedding matrix. Then we have the following definitions:
>
> - Whitening Transformation: $(H^{\top}H)^{-1/2}=VS^{-1}V^{\top}$.
>
> - Whitened Embedding: $(H^{\top}H)^{-1/2}H^{\top}=VU^{\top}$, which removes the scales of singular values.
>
> - Moore-Penrose pseudoinverse: $(H^{\top}H)^{-1}H^{\top}=VS^{-1}U^{\top}$, which inverses the singular values.
>
> The above two equations have close connections:
>
> - As the original paper explained, Moore-Penrose pseudoinverse consists of both whitening transformation and whitened embedding: $\left(H_I^{\top} H_I\right)^{-1}H_I^{\top} = \underbrace{\left(H_I^{\top} H_I\right)^{-1/2}}\_{\textbf{Whitening matrix}} \quad \underbrace{\left(H_I^{\top} H_I\right)^{-1/2}H_I^{\top}}\_{\textbf{Whitened embeddings}}$, implying that matching the Moore-Penrose pseudoinverse can also learn the information about matrix whitening.
>
> - The covariance $H^{\top}H$ is low-rank, meaning that there are a lot of zero singular values. When calculating the pseudoinverse, we often use $H^{\top}H + \alpha I$ to prevent numerical instability. Therefore, $(S+\alpha I)^{-1} \approx \frac{1}{\alpha}I$.
>
> &nbsp;
>
> > **Q2: Comparison between Eq. 5 and Eq. 7**
>
> A5: We denote Eq. 5 as **Raw Version** and Eq. 7 as **Stable Version** to clearly elaborate their connections.
>
> Compared to the raw version, the stable version considers three additional issues: Embedding Shift (ES), Scale Explosion (SE), and Matrix Inversion (MI), and modifies the raw vision to stabilize the distillation process, while only slightly increasing the time complexity.
>
> - **Complexity.** Taking $\Sigma_{II}$ as an example:
>     - Raw Version: $\Sigma_{II}=H_I^{\top} H_I$
>     - Stable Version: $\Sigma_{II}= \underbrace{\frac{1}{|\mathcal D|}}\_{SE} \underbrace{(H_I-\mu_I)^{\top}}\_{ES}(H_I-\mu_I)+ \underbrace{\alpha I}_{MI}$
>     - The additional complexity mainly comes from the calculation of the mean value $\mu_I$ and the addition of the identity matrix, which is acceptable.
>
> - **Ablation Study.** We further make an ablation study on the Flickr-30k dataset to validate the effectiveness of the stable version. The results are shown below. We have the following observations:
>
>     - Removing ES slightly affects the performance of APM. The reason is that synthetic data is initialized by real data, and they may have similar mean values.
>
>     - Removing SE significantly reduces the retrieval performance as the number of real data is larger than the synthetic data, making the scale of $H_I^{\top}H_I$ and $\hat{H}_I^{\top}\hat{H}_I$ different.
>
>     - Removing MI cannot obtain a meaningful synthetic dataset. The covariance of the synthetic dataset is low-rank, and directly solving for its inverse matrix will lead to numerical instability.
>
> |                  | IR@1 | IR@5 | IR@10 | TR@1 | TR@5 | TR@10 |
> |------------------|------|------|-------|------|------|-------|
> | $\mathcal{L}_{\text{APM}}$ | 12.8 | 34.2 | 47.1 | 17.8 | 43.0 | 57.2 |
> | w/o ES           | 11.6 | 32.0 | 44.9 | 17.3 | 41.8 | 57.5 |
> | w/o SE           | 2.1  | 7.5  | 12.9 | 3.9  | 13.3 | 19.9 |
> | w/o MI           | 0.2  | 0.8  | 1.3  | 0.0  | 0.0  | 0.0  |

---

> ### Comment · Reviewer_kpau · 2025-11-25
> **Official Comment by Reviewer kpau**
>
> Thank you for addressing some of my concerns, but you haven't effectively resolved my question about why the least squares objective function is used. Therefore, I will maintain my initial score.

---

> > ### Author Response · Authors · 2025-12-03
> > **Response to Reviewer kpau (2/2)**
> >
> > The reasons for using the least squares objective function, i.e., $||UV^{\top}-I||$, as a proxy of InfoNCE are two-fold:
> >
> > - Both loss functions aim to maximize the similarity between positive pairs and minimize the similarity between negative pairs.
> >
> > - According to the general response, the least squares objective function has similar analytic projector parameters to the InfoNCE loss.
> >
> > Based on the two points, the least squares objective function is a valid proxy of InfoNCE loss.

---

### Official Review · Reviewer_qAEJ · 2025-10-29

**Soundness:** 2
**Presentation:** 3
**Contribution:** 2
**Rating:** 4
**Confidence:** 2

**Summary:**

This paper introduces APM, a novel and efficient framework for multi-modal dataset distillation. Unlike prior trajectory-matching-based MDD methods that require storing many model checkpoints and performing costly double backpropagation, APM directly aligns analytic parameters of linear modal projectors between real and synthetic datasets. This approach eliminates the need for trajectory storage and bi-level optimization, significantly improving scalability and efficiency.

**Strengths:**

1. The analytic parameter matching formulation is a creative alternative to trajectory matching, offering a new theoretical and algorithmic perspective on MDD by connecting it with matrix whitening.
2. APM addresses major scalability and efficiency bottlenecks in multi-modal dataset distillation, achieving substantial computational and storage reductions while maintaining or improving performance. This has high practical relevance for large-scale multimodal learning.

**Weaknesses:**

1. Main concern: APM assumes linear modal projectors (e.g., CLIP-style architectures). Extending it to nonlinear or generative models remains an open challenge, which slightly limits its general applicability.
2. While results on Flickr30k and MS-COCO are strong, additional evaluation on more diverse multi-modal domains (e.g., video–text or audio–text) could better demonstrate generality.
3. The paper could provide deeper analysis of why APM performs better under small budgets, and explore whether whitening-based isotropy fully explains the performance gains.

**Questions:**

see the weaknesses.

---

> ### Author Response · Authors · 2025-11-21
> **Response to Reviewer qAEJ**
>
> We are grateful for your constructive advice and the opportunity to address your concerns.
>
> &nbsp;
>
> > **W1: Extending linear projectors**
>
> A1: Please see the general response.
>
> &nbsp;
>
> > **W2: Evaluation on more diverse multi-modal domains**
>
> A2: We tend to explore our model on the audio-text dataset. Currently, only Repblend [1] conducts experiments on the AudioCaps dataset, which uses EfficientAT as the audio encoder and BERT as the text encoder.
>
> The code for Repblend on the audio-text dataset is not publicly available, and the paper does not describe its implementation in detail.
>
> We make a lot of efforts to reproduce their performance on the full dataset (AR@1=21.3, AR@5=53.2, AR@10=68.5; TR@1=25.2, TR@5=58.8, TR@10=71.6). However, the reproduced performance is far from that reported.
>
> As the rebuttal time is approaching, we cannot evaluate the performance of APM on more diverse domains. Nevertheless, we should still emphasize the generalizability of APM. The calculation of analytic parameters relies on the data embeddings rather than a certain modality. Therefore, it can be used for arbitrary multi-modal dataset distillation.
>
> [1] Beyond Modality Collapse: Representations Blending for Multimodal Dataset Distillation. NeurIPS 2025.
>
> &nbsp;
>
> > **W3: Performance of APM and whitening-based isotropy**
>
> A3: We first need to clarify that APM performs well on both small and large budgets. For example, in the MS-COCO dataset, APM consistently outperforms baselines under 100/500 pairs:
>
> | 100-pairs | IR@1 | IR@5 | IR@10 | TR@1 | TR@5 | TR@10 |
> |---|------|------|-------|------|------|-------|
> | LoRS | 1.8 | 7.1 | 12.2 | 3.3 | 12.2 | 19.6 |
> | RepBlend | 4.1 | 13.9 | 22.3 | 5.2 | 17.9 | 28.0 |
> | APM | **4.7**  | **16.2** | **25.8** | **6.2** | **20.0** | **31.1** |
>
> | 500-pairs | IR@1 | IR@5 | IR@10 | TR@1 | TR@5 | TR@10 |
> |---|------|------|-------|------|------|-------|
> | LoRS | 2.8 | 9.9 | 16.5 | 5.3 | 18.3 | 27.9 |
> | RepBlend | 6.2 | 19.9 | 30.6 | 7.0 | 22.0 | 32.9 |
> | APM | **7.1**  | **21.8** | **33.3** | **8.0** | **24.3** | **37.1** |
>
> In the Flickr-30k dataset, APM performs well in the Image-retrieval (IR) and is slightly outperformed by RepBlend in Text-retrieval (TR).
> We attribute this to the caption synthesis, which is optimized in the embedding space and projected to existing captions, limiting the flexibility of APM.
>
>
> | 500-pairs | IR@1 | IR@5 | IR@10 | TR@1 | TR@5 | TR@10 |
> |---|------|------|-------|------|------|-------|
> | LoRS | 10.0 | 28.9 | 41.6 | 15.5 | 39.8 | 53.7 |
> | RepBlend | 17.0 | 42.5 | 55.9 | **22.5** | **53.2** | **66.7** |
> | APM | **17.5**  | **43.5** | **56.8** | 21.6 | 52.7 | 66.4 |
>
>
> Next, we make an experiment to verify the effectiveness of isotropic distribution.
>
> As explained in lines 182-190, the analytic parameters $(H_I^{\top}H_I)^{-1}H_I^{\top}V(V^{\top}V)^{-1}$ have a close connection with matrix whitening, which transforms the embeddings into an isotropic distribution.
>
> To construct an anisotropic distribution, we remove the inverse covariance in the analytic parameters and obtain $H_I^{\top}V$, which is dominated by the principal singular values.
>
> The comparison between the isotropy and anisotropy distributions is shown below. We have the following observations:
>
> - The synthetic data learned by an isotropic distribution has large entropy, indicating that it can encode more information about the real data.
>
> - The isotropic distribution outperforms the anisotropic distribution by a large margin, verifying the effectiveness of the isotropic distribution and supporting our claims.
>
> | | Equation | Img_Entropy | Text_Entropy | IR@1 | TR@1 |
> |---|---|---|---|---|---|
> | Isotropic | $(H_I^{\top}H_I)^{-1}H_I^{\top}V(V^{\top}V)^{-1}$ | **3.93** | **3.87** | **12.8** | **17.8** |
> | Anisotropic | $H_I^{\top}V$ | 3.87 | 3.79 | 7.9 | 11.2 |

---

> > ### Comment · Reviewer_qAEJ · 2025-11-25
> >
> > Thank you for the reply. I would like to keep my score for now as the evaluation on multi-modal domains is limited.

---

> > > ### Author Response · Authors · 2025-12-03
> > > **Response to Reviewer qAEJ (2/2)**
> > >
> > > With a lot of effort, we finally extended our method to the audio-text datasets. We have added all experimental details in the revision. Hope to address your concerns.
> > >
> > > **To improve the reproducibility, we introduce the implementation details and release our code and logs in [anonymous github](https://anonymous.4open.science/r/ICLR-735-MMDD)**
> > >
> > > Following RepBlend [1], we choose the [AudioCaps](https://github.com/cdjkim/audiocaps/tree/master/dataset) dataset, consisting of 49,838 training audios, 495 validation audios, and 975 test audios. We use [EfficientAT (mn20\_as)](https://github.com/fschmid56/EfficientAT) as the audio encoder and BERT as the text encoder.
> > >
> > > - **Data Preparation.** The AudioCaps dataset contains files in WAV format. We use the ``AugmentMelSTFT`` function from EfficientAT to preprocess the audio. We sample the audios in mono at a sampling rate of 32 kHz and then calculate their log-mel-spectrogram in a 25-ms window with a step size of 10 ms. After processing, each audio has a feature map with shape [1, 128, 1000].
> > >
> > > - **Distillation.** Instead of distilling the raw audios, we directly synthesize the log-mel-spectrogram to match the input of EfficientAT. The spectrogram has a shape of [1, 128, 1000], which can be seen as an image with channel=1, width=128, and height=1000. Therefore, the code of image-text distillation can be directly transferred to the audio-text datasets.
> > >
> > > - **Evaluation.** In the test set of AudioCaps, each audio corresponds to 5 captions, which improves its retrieval performance. We train the multi-modal network from scratch and evaluate it on the test set. We repeat the experiments five times and report the average performance and standard deviation.
> > >
> > > - **Results.** The results are shown below. We can observe that APM outperforms LoRS and RepBlend, especially in audio retrieval tasks, where it shows a significant performance improvement. The visualizations of real and synthetic data can be seen in Figure 6 in the revision.
> > >
> > >
> > > Results on AudioCaps dataset. We use EfficientAT (mn20\_as)+BERT as the distillation and evaluation networks. Full dataset performance: AR@1=17.6, AR@5=47.7, AR@10=63.8; TR@1=20.6, TR@5=49.6, TR@10=67.2. The best results are highlighted in bold.
> > >
> > > | Pairs | Method   | AR@1        | AR@5         | AR@10        | TR@1         | TR@5         | TR@10        |
> > > | ----- | -------- | ----------- | ------------ | ------------ | ------------ | ------------ | ------------ |
> > > | 100   | LoRS     | 2.7±0.3     | 8.6±0.3      | 14.7±0.4     | 5.9±0.3      | 13.0±0.4     | 21.8±0.5     |
> > > | 100   | RepBlend | 4.1±0.2     | 14.2±0.3     | 23.7±0.4     | 8.9±0.1      | 24.3±0.2     | 34.7±0.3     |
> > > | 100   | APM      | **8.3±0.3** | **28.6±0.3** | **42.1±0.4** | **11.3±0.6** | **33.4±0.6** | **46.7±0.8** |
> > > | 200   | LoRS     | 3.8±0.2      | 14.8±0.2     | 21.8±0.2     | 8.0±0.2      | 21.2±0.2     | 33.1±0.2     |
> > > | 200   | RepBlend | 6.8±0.2      | 20.6±0.2     | 31.4±0.3     | 9.7±0.2      | 29.1±0.4     | 41.2±0.4     |
> > > | 200   | APM      | **10.1±0.1** | **32.5±0.3** | **47.3±0.2** | **11.7±0.7** | **35.6±0.8** | **51.1±1.1** |
> > > | 500   | LoRS     | 7.1±0.1      | 24.7±0.2     | 36.7±0.2     | 9.2±0.2      | 27.4±0.3     | 41.3±0.3     |
> > > | 500   | RepBlend | 9.7±0.1      | 32.2±0.3     | 46.8±0.2     | **13.8±0.3** | 38.6±0.3     | 54.1±0.4     |
> > > | 500   | APM      | **11.4±0.1** | **35.8±0.4** | **51.3±0.3** | 13.6±0.7     | **39.3±0.7** | **54.8±0.5** |

---

### Official Review · Reviewer_q6bY · 2025-10-30

**Soundness:** 3
**Presentation:** 3
**Contribution:** 3
**Rating:** 6
**Confidence:** 4

**Summary:**

The paper introduces a novel Analytic Parameter Matching (APM) framework for multi-modal dataset distillation, which replaces traditional bi-trajectory matching with a direct analytic alignment of modal projectors. This approach is both original and practical, as it eliminates the need for storing multiple checkpoints and performing double backpropagation, thereby achieving substantial gains in computational and storage efficiency. The approach is theoretically grounded, as the authors derive a closed-form solution for linear projectors and demonstrate its equivalence to matrix whitening, offering a clear statistical interpretation of the method.

**Strengths:**

* The method is theoretically well-grounded: the authors derive a closed-form solution for linear projectors and establish its equivalence to matrix whitening, offering a clear statistical interpretation of the proposed formulation.

* Empirically, APM demonstrates strong performance and scalability across Flickr30k and MS-COCO, achieving up to 65$\times$ reduction in storage and 9.6$\times$ speed-up compared to the previous work while maintaining competitive retrieval accuracy. Overall, the work combines theoretical clarity, implementation simplicity, and practical significance, providing a meaningful advance toward efficient multi-modal data distillation.

**Weaknesses:**

* While the proposed framework is elegant and efficient, it relies heavily on the assumption of linear modal projectors. This restricts its applicability to modern multi-modal models that employ non-linear or attention-based projection mechanisms.
* The comparison is limited to trajectory-based distillation baselines such as LoRS and RepBlend. The paper omits recent generative distillation approaches, notably EDGE [1], which also address efficiency and scalability through generative priors.

[1] Zhao, Zhenghao, et al. "Efficient Multimodal Dataset Distillation via Generative Models." arXiv e-prints (2025): arXiv-2509.

**Questions:**

* (with W1)
Since the analytic formulation assumes linear projectors, could the authors discuss whether APM can generalize to a simple non-linear setting, for example, one involving a single activation layer?
* (with W2)
As EDGE also demonstrates an efficient distillation process, could the authors provide an explicit comparison between EDGE and APM in Tables 1 and 2?
* The evaluation currently focuses exclusively on cross-modal retrieval. It would be informative to test APM-distilled datasets on other downstream tasks, such as VQA or zero-shot classification on different datasets, to further assess their semantic generalization capability.

---

> ### Author Response · Authors · 2025-11-21
> **Response to Reviewer q6bY**
>
> We appreciate your detailed review and the recognition of our contributions.
>
> &nbsp;
>
> > **W1 & Q1: Non-linear projector**
>
> A1: See the general response.
>
> &nbsp;
>
> > **W2 & Q2: Comparision between EDGE and APM**
>
> A2: EDGE leverages generative models to address the semantic correlation and diversity issues of existing MDD methods. Notably, we have cited EDGE in the original paper and will add its results to the quantitative experiments.
>
> EDGE mainly focuses on the large budget setting (500 / 1000 pairs). We report the performance of EDGE and APM under the same settings in the revision. Below is a quick comparison.
>
> | Flickr | IR@1 | IR@5 | IR@10 | TR@1 | TR@5 | TR@10 |
> |---|------|------|-------|------|------|-------|
> | EDGE-500 | 6.7 | 21.0 | 30.5 | 13.3 | 35.6 | 47.5 |
> | APM-500 | **17.5** | **43.5** | **56.8** | **21.6** | **52.7** | **66.4** |
> | EDGE-1000 | 9.9 | 28.2 | 40.5 | 14.5 | 38.3 | 51.7 |
> | APM-1000 | **18.4** | **45.5** | **57.9** | **23.2** | **53.8** | **66.9** |
>
> | MS-COCO | IR@1 | IR@5 | IR@10 | TR@1 | TR@5 | TR@10 |
> |---|------|------|-------|------|------|-------|
> | EDGE-500 | 1.8 | 6.5 | 11.2 | 2.9 | 9.5 | 15.7 |
> | APM-500 | **7.1** | **21.8** | **33.3** | **8.0** | **24.3** | **37.1** |
>
>
> &nbsp;
>
> > **Q3: Exploration on downstream tasks**
>
> A3: To verify whether the synthetic dataset can be used in downstream tasks beyond retrieval. We make a zero-shot image classification task to benchmark the performance between real and synthetic datasets.
>
> Specifically, we use three datasets, CIFAR-10, CIFAR-100, and ImageNet-1k. The results are shown below. We can see that the synthetic datasets have similar zero-shot classification performance to the real dataset.
>
> |Dataset|CIFAR-10 Top-1(%)|CIFAR-10 Top-5(%)|CIFAR-100 Top-1(%)|CIFAR-100 Top-5(%)|ImageNet-1k Top-1(%)|ImageNet-1k Top-5(%)|
> |---|---|---|---|---|---|---|
> |Syn-99|52.44|87.54|13.53|32.19|4.28|12.50|
> |Syn-199|53.73|85.96|13.70|34.76|4.35|12.69|
> |Syn-499|**55.03**|**90.28**|**14.45**|**36.14**|**5.06**|**14.77**|
> |Full|58.77|92.07|16.34|38.27|7.62|19.54|

---

### Official Review · Reviewer_teJU · 2025-11-01

**Soundness:** 3
**Presentation:** 1
**Contribution:** 3
**Rating:** 4
**Confidence:** 4

**Summary:**

This paper proposes an efficient method for multimodal dataset distillation by bypassing double backpropagation computation. This was achieved by leveraging an analytic solution of the image/text projection head of the vision-language model.

**Strengths:**

- Simple, novel, and reasonable method
  - The proposed method is conceptually simple and novel, and nicely addresses the efficiency issue of existing methods by leveraging an analytic solution.
- Insightful experiments
  - The authors provide experiments on the standard benchmark suite as well as interesting analysis on SVD entropy, trying to validate their method on multiple perspectives.

**Weaknesses:**

- Reader-unfriendly presentation and weak logical flow of writing
  - Figure 1 does not help in understanding the essence of trajectory matching and the proposed method. Even after fully reading this paper and understanding the proposed method, I don't think this figure helps a smooth introduction to naive readers. It seems like the authors' visualization was motivated by that of Figure 2 in Wu et al. 2024, but I think that visualization from Wu et al. 2024 is also not that informative. It would be better to add more annotations, as done in Figure 3 of Cazenavette et al. 2022. And the author should emphasize how their method is different from the trajectory matching.
  - In line 073, the authors mentioned the double backpropagation issue of the previous method without any short description of what it means. Since the core contribution of this work is addressing that problem, I think the authors should elaborate on what the double backpropagation is from the introduction (at least briefly). Although they mentioned that in L134 again, the detail is missing -- please elaborate on one backprop for what, and another one for what.
  - In line 144, "propose to align the optimal parameters" align between which parameters? -- It would be better to explicitly spell out (e.g., align the optimal parameters of the model trained on real and the model trained on synthetic).
  - In line 175, "cosine similarity for searching, which requires an isotropic distribution" -- this is not true. Cosine similarity-based retrieval itself does not require anything, but the authors say as if it is a necessary property.
  - In line 187, "As the modal projectors contain whitened embeddings, ..., as a surrogate of MDD."
    - Since the modal projectors are just conceptual components, the authors should specify them further, like "As the optimal solution parameters of the modal projectors contain ~"
    - a surrogate of MDD? This is also a very imprecise expression --> Surrogate of [XXX] in MDD would be a better expression where [XXX] can be the entire model parameters, something like that.
  - In Section 3.2, the authors point out that the derived analytic solution in Eq. (5) does not truly achieve the whitening due to the lack of zero mean centering. Then, why do they pretend that it is whitening in Eqs (5) and (6)? It would be better the carefully mention this subtle difference in advance.
- Limited scope of validation, their effectiveness, and reliability
  - As the authors mentioned in L107, the goal of dataset distillation is to achieve comparable performance to the original dataset with far fewer samples. However, they do not provide a comparison with the full dataset results done in Cazenavette et al. 2022.
  - This makes it hard to infer how significantly the proposed method reduces the performance gap between the existing methods and the upper bound method (full dataset).
  - It is worth noting that the authors borrow performance metrics of baseline methods in Table 7 (scalability experiments with 1000 and 2000 data pairs) from a previous work.
  - Compared to the performance obtained from the 500 data pairs in Table 2, which the authors might reproduce themselves, 1000 pairs and 2000 pairs cases in Table 7 show poorer performance of LoRS. Therefore, the reliability of Table 7 results is questionable.
- Lack of discussion on the methodology design
  - The authors made a lot of tweaks to derive their loss $\mathcal{L}_{APM}$ from the true analytic solution in Eq. (5).
  - However, they do not discuss how this tweak makes the proposed final loss term $\mathcal{L}_{APM}$ deviate from the original analytic solution, and why it is still valid to be used as a proxy of an iterative optimization-based solution.
- Lack of discussion on the observed performance
  - In Table 2 at a 500-pairs setup, the proposed method underperforms a competitive baseline, but the authors do not provide a detailed discussion on why their proposed method shows limited data size scalability compared to RepBlend.
  - As Table 7 does not contain RepBlend, I am further speculating on the scalability of APM compared to RepBlend.


---

> Reference

- Cazenavette et al. 2022. "Dataset Distillation by Matching Training Trajectories"
- Wu et al. 2024. "Vision-Language Dataset Distillation"

**Questions:**

See the weaknesses section, please.

If there is any misunderstanding from me, feel free to point out.

---

> ### Author Response · Authors · 2025-11-21
> **Response to Reviewer teJU**
>
> Thanks for the detailed and helpful comments. We reply to the comments in detail. Hope to address your concerns.
>
> &nbsp;
>
> > **W1: Paper Presentation**
>
> A1: We respond to the raised questions point by point.
>
> - We replace Figure 1 in the revision to illustrate our idea better.
>
> - We explained its meaning in lines 47-49, where double backpropagation first appears. "TM involves double backpropagation during distillation, which first updates the model parameters and then optimizes the synthetic dataset by minimizing the differences between expert and student trajectories."
>
> - We revise the description in line 144. "we propose to align the optimal parameters of the modal projectors trained on real and synthetic datasets"
>
> - The statement about cosine similarity is to emphasize the importance of isotropic distribution. As cosine similarity uses L2-normalization to remove the scale information, the isotropic embedding may be more suitable for cosine similarity in retrieval task. We revise this to "Existing MDD methods focus on the retrieval task, isotropic distributions~\citep{BERT_whitening} are more suitable for cosine similarity-based searches." in lines 186-187.
>
> - We revise the statement of the surrogate to "As the optimal parameters of the modal projectors contain whitened embeddings, it is reasonable to use them as a surrogate for the entire model parameters." in lines 198-200.
>
> &nbsp;
>
> > **W2: Evaluation of APM**
>
> A2.1: Notably, MTT (Cazenavette et al. 2022) is used for image distillation rather than multi-modal dataset. Hence, we cannot directly compare it with APM.
>
> For the full dataset performance, we have reported it in the captions of Tables 2 and 3.
>
> | Flickr | IR@1 | IR@5 | IR@10 | TR@1 | TR@5 | TR@10 |
> |---|------|------|-------|------|------|-------|
> | Full | 21.3 | 51.0 | 63.6 | 31.1 | 61.7 | 74.3 |
> | APM-500 | 17.5 | 43.5 | 56.8 | 21.6 | 52.7 | 66.4 |
>
> | MS-COCO | IR@1 | IR@5 | IR@10 | TR@1 | TR@5 | TR@10 |
> |---|------|------|-------|------|------|-------|
> | Full | 11.1 | 31.5 | 44.7 | 14.6 | 37.6 | 50.5 |
> | APM-500 | 7.1 | 21.8 | 33.3 | 8.0 | 24.3 | 37.1 |
>
> &nbsp;
>
> A2.2: For the budget of 1000 and 2000 pairs, we directly use the performance from MDW [1]. We also notice that the performance of LoRS is lower than 500-pairs. To make a fair comparison, we reproduce the results of LoRS and RepBlend and report them below:
>
> | Flickr | IR@1 | IR@5 | IR@10 | TR@1 | TR@5 | TR@10 |
> |---|------|------|-------|------|------|-------|
> | LoRS-1000 | 11.0 | 30.8 | 42.5 | 16.0 | 41.1 | 54.8 |
> | RepBlend-1000 | 17.8 | 44.7 | 56.9 | 23.0 | **54.4** | **67.3** |
> | APM-1000 | **18.4** | **45.5** | **57.9** | **23.2** | 53.8 | 66.9 |
>
> In the revision, we will replace Table 7 with this new table.
>
> [1] Multi-modal dataset distillation in the wild.
>
> &nbsp;
>
> > **W3: Modification of the analytic parameters**
>
> A3: We denote Eq. 5 as **Raw Version** of analytic parameters and $\mathcal{L}_{\text{APM}}$ as **Stable Version** to clearly elaborate their connections.
>
> - Raw Version: $(H_I^{\top}H_I)^{-1}H_I^{\top}V(V^{\top}V)^{-1}$
>
> - Stable Version: $\Sigma_{II}^{-1}\Sigma_{IV}\Sigma_{VV}^{-1}$, where $\Sigma_{II}= \frac{1}{|\mathcal D|} (H_I-\mu_I)^{\top}(H_I-\mu_I)+ \alpha I$.
>
> Compared to the raw version, the stable version adds three operators: Zero-centering ($H_I - \mu_I$), Scale Normalization ($\frac{1}{|\mathcal D|}$), and Inverse Stability ($\alpha I$), corresponding to the Embedding Shift (ES), Scale Explosion (SE), and Matrix Inversion (MI) issues in Eq. 5.
>
> These operators do not change the essence of analytic parameters but stabilize the distillation process. Below is an ablation study.
>
> || IR@1 | IR@5 | IR@10 | TR@1 | TR@5 | TR@10 |
> |---|------|------|-------|------|------|-------|
> | $\mathcal{L}_{\text{APM}}$ | 12.8 | 34.2 | 47.1 | 17.8 | 43.0 | 57.2 |
> | w/o ES           | 11.6 | 32.0 | 44.9 | 17.3 | 41.8 | 57.5 |
> | w/o SE           | 2.1  | 7.5  | 12.9 | 3.9  | 13.3 | 19.9 |
> | w/o MI           | 0.2  | 0.8  | 1.3  | 0.0  | 0.0  | 0.0  |
>
> &nbsp;
>
> > **W4: Observed Performance**
>
> A4: We provide a new Table 7 to show the performance of LoRS, RepBlend, and APM under 1000 pairs.

---

> ### Comment · Reviewer_teJU · 2025-11-28
>
> I really appreciate the authors' professional rebuttal. I think the quality of the draft has been fairly improved compared to the original version. Still, I think the performance gain compared to RepBlend is marginal somehow. But anyway, there is a consistent improvement by the proposed method.
>
> (minor: in Table 3, the Pairs column has a typo in its percentage expression. Please fix them)
>
> I want to revise my initial review scores (presentation 1 -> 3) and maybe the overall recommendation 4->6 as well, but the system right now does not show the "edit review" button. After this bug is fixed, I am planning to edit my review.

---

> > ### Author Response · Authors · 2025-12-03
> > **Response to Reviewer teJU (2/2)**
> >
> > We are very grateful for your valuable suggestions and appreciation of our paper.
> >
> > We kindly clarify that the percent sign (%) in Table 2 denotes parts per hundred, whereas the per mille sign (‰) in Table 3 denotes parts per thousand.

---

### Official Review · Reviewer_P2Ji · 2025-11-02

**Soundness:** 3
**Presentation:** 3
**Contribution:** 3
**Rating:** 6
**Confidence:** 4

**Summary:**

This paper proposes Analytic Parameter Matching (APM), a new framework for efficient multi-modal dataset distillation (MDD). Instead of aligning entire optimization trajectories between expert and student models, APM directly matches the analytic parameters of linear modal projectors in CLIP-style models. This design removes the need for trajectory storage and double backpropagation, cutting both storage and computation costs. The authors connect these analytic parameters to matrix whitening, showing they improve the alignment across modalities. Empirically, APM achieves large efficiency gains while maintaining or improving performance over prior MDD methods on benchmarks like Flickr30k and MS-COCO.

**Strengths:**

- The proposed APM method reduces both computational and storage costs by eliminating trajectory storage and double backpropagation, achieving impressive speedup and compression ratios.

- The authors provide an analytic formulation linking APM to matrix whitening, offering intuitive insight into why the method improves cross-modal isotropy and alignment. APM achieves strong empirical results across multiple datasets and model architectures.

- The method scales to larger synthetic datasets and demonstrates strong cross-architecture generalization, highlighting robustness and practicality for real-world multi-modal distillation tasks.

**Weaknesses:**

- The method primarily focuses on linear projectors e.g., CLIP-style models, which may limit its applicability to architectures with non-linear projection heads or more complex fusion mechanisms.

- The experiments are limited to cross-modal retrieval; the work does not explore other multi-modal downstream tasks such as captioning to test broader generalization.

- The paper could provide more discussion on potential trade-offs when applying to more complex datasets.

**Questions:**

How well does APM generalize to multi-modal models with non-linear or transformer-based projection heads, where analytic parameter computation may not be straightforward?

---

> ### Author Response · Authors · 2025-11-21
> **Response to Reviewer P2Ji**
>
> &nbsp;
>
> > **W1 & Q1: Non-linear projector**
>
> A1: Please see the general response.
>
> &nbsp;
>
> > **W2: Broader generalization**
>
> A2: To verify whether the synthetic dataset can be used in downstream tasks beyond retrieval. We make a zero-shot image classification task to benchmark the performance between real and synthetic datasets.
>
> Specifically, we use three datasets, CIFAR-10, CIFAR-100, and ImageNet-1k. The results are shown below. We can see that the synthetic datasets have similar zero-shot classification performance to the real dataset.
>
> |Dataset|CIFAR-10 Top-1(%)|CIFAR-10 Top-5(%)|CIFAR-100 Top-1(%)|CIFAR-100 Top-5(%)|ImageNet-1k Top-1(%)|ImageNet-1k Top-5(%)|
> |---|---|---|---|---|---|---|
> |Syn-99|52.44|87.54|13.53|32.19|4.28|12.50|
> |Syn-199|53.73|85.96|13.70|34.76|4.35|12.69|
> |Syn-499|**55.03**|**90.28**|**14.45**|**36.14**|**5.06**|**14.77**|
> |Full|58.77|92.07|16.34|38.27|7.62|19.54|
>
> For the more challenging generation task, e.g., captioning, existing MDD datasets are not suitable as they are trained on a classification model, which is hard to generalize to generation tasks.
>
> &nbsp;
>
> > **W3: Discussion on potential trade-offs when applying to more complex datasets**
>
> A3: For more complex datasets, APM may face a natural trade-off: while analytic matching remains highly efficient, complex datasets often exhibit higher intra-class variance and multi-modal distributions, making the linear whitening assumption less ideal. In such cases, APM may require slightly larger distilled sets or introduce the non-linear projectors to increase its expressive power. We will add this clarification in the revision.
>
> &nbsp;

---

### Author Response · Authors · 2025-11-21
**General Response from Authors**

We thank all reviewers for the valuable feedback and suggestions. We have addressed most of the questions and suggestions in the revised paper (highlighted in blue).

We respond to the common concerns here and reply to individual reviewers for other questions.

&nbsp;

> **Exploring non-linearity in APM**

In the revision, we provide a new theoretical analysis of the non-linearity in multi-modal contrastive learning. We give the detailed proofs in **Appendix B** and briefly introduce the conclusions below.

&nbsp;

``Conclusion 1: InfoNCE loss with softmax activation has similar analytic solutions with regression loss``

InfoNCE loss is widely used in multi-modal contrastive learning, and we simplify it into a regression loss $||UV^{\top} - I||$, which has analytic solutions. Now, we can also obtain the analytic parameters of InfoNCE. Below is the derivation process.

- Previous literature [1, 2] gives the analytic solution of a linear layer with softmax activation in the multi-class classification task. Specifically, the probability that a sample belongs to a certain class is defined as: $p(i|x) = \frac{\exp(xw_i^{\top} + b_i)}{\sum_{i=1}^{k} \exp(xw_i^{\top} + b_i)},$ where $x$ is the sample, and $w_i$ and $b_i$ denote the weight and bias of the $i$-th class, respectively. The analytic solutions of $w_i$ and $b_i$ are defined as:
$$w_i = \mu_i \Sigma^{-1}, \quad b_i = {\ln p_i} - \frac{1}{2}\mu_i \Sigma^{-1} \mu_i^{\top},$$ where $p_i$ is the ratio of the $i$-th class, $\mu_i$ is the mean value of the data embeddings in the $i$-th class, and $\Sigma = \hat{\Sigma} + \hat{\mu}^{\top}\hat{\mu} + \sum_i p_i \mu_i^{\top} \mu_i.$ ($\hat{\mu}$ and $\hat{\Sigma}$ denote the mean and covariance of $x$)

- The InfoNCE loss $\mathcal{L}\_{\text{NCE}}= - \sum_i \log \frac{\exp(u_i v_i^{\top}/\tau)}{\sum_{j=1}^{M} \exp(u_i v_j^{\top}/\tau)}$ can also be viewed as a multi-class classification function, where each pair is a class. In this case, **$u_i$ can be seen as the sample and $v_i^{\top}$ denotes the weight in the $i$-th class.** Therefore, we can directly obtain their analytic solutions:
$$U^* = \frac{1}{\tau} V \Sigma_V^{-1}=\frac{N}{2\tau}V(V^{\top}V)^{-1}, \quad V^* = \frac{1}{\tau} U \Sigma_U^{-1} = \frac{N}{2\tau} U(U^{\top}U)^{-1}.$$

- On the other hand, the analytic solution of the regression loss is defined as:
$$U^* = V(V^{\top}V)^{-1}, \quad V^* = U(U^{\top}U)^{-1}.$$

We can observe that the analytic parameters of InfoNCE and regression loss are similar, the difference being a scaling constant $\frac{N}{2\tau}$.

&nbsp;

``Conclusion 2: Non-linear projectors also have analytic parameters``

For the non-linear projectors $U=\sigma(H_IW_I)$ and $V=\sigma(H_TW_T)$, we have $\sigma(H_IW_I)=V(V^{\top}V)^{-1}$ and $\sigma(H_TW_T)=U(U^{\top}U)^{-1}$.

Since we have obtained the optimal results of $U$ and $V$, and the activation $\sigma$ is an element-wise function, we can directly use its inversion function to calculate the analytic parameters:
$$W_I^* = (H_I^{\top}H_I)^{-1}H_I^{\top} \sigma^{-1}(V(V^{\top}V)^{-1}), \quad W_T^* = (H_T^{\top}H_T)^{-1}H_T^{\top} \sigma^{-1}(U(U^{\top}U)^{-1}).$$

We list some commonly used activation functions and their inverses below.

| Activation |  $\sigma(x)$ | $\sigma^{-1}(y)$ |
|-----------|--------------|------------------|
| **Sigmoid** | $$\sigma(x)=\frac{1}{1+e^{-x}}$$ | $$\sigma^{-1}(y)=\ln \left(\frac{y}{1-y}\right)$$ |
| **Tanh** | $$\sigma(x)=\tanh(x)=\frac{e^{x}-e^{-x}}{e^{x}+e^{-x}}$$ | $$\sigma^{-1}(y)=\operatorname{arctanh}(y)=\frac{1}{2}\ln \left(\frac{1+y}{1-y}\right)$$ |
| **LeakyReLU** | $$\sigma(x)=\begin{cases}x, & x\ge 0 \\\\ \alpha x, & x<0\end{cases}$$ | $$\sigma^{-1}(y)=\begin{cases}y, & y\ge 0 \\\\ y/\alpha, & y<0\end{cases}$$ |

[1] [Does logistic regression have an analytical solution?](https://spaces.ac.cn/archives/8578)

[2] [Easy Logistic Regression with an Analytical Solution](https://towardsdatascience.com/easy-logistic-regression-with-an-analytical-solution-eb4589c2cd2d/)

---

### Author Response · Authors · 2025-12-03
**Summary of Rebuttal and Discussion**

Dear AC,

We sincerely thank you for your efforts under such special circumstances. To reduce your workload, we briefly summarize the questions raised by reviewers in the rebuttal stage and our responses to them.

&nbsp;

**[Exploring Non-linearity in APM (All Reviewers)]**

In the [general response](https://openreview.net/forum?id=Fxz0aaGSNY&noteId=4wIxe5gIbs), we provide two additional theoretical analyses:
- InfoNCE loss (with softmax) has similar analytic projector parameters to the regression loss.
- Non-linear projectors can calculate analytic parameters based on inverse activation functions.

The new theoretical insights extend the proposed Analytical Parameter Matching (APM) method from the linear case to a wider range of nonlinear cases, improving the generalization and practicality of APM.

**[Broader Downstream Tasks (P2Ji & q6bY)]**

We add an experiment on the zero-shot image classification tasks to verify the generalizability of APM. The results show that APM can achieve similar zero-shot performance with the full dataset.

**[Modification of Analytic Parameters (teJU & kpau)]**

We provide three modifications to stabilize the distillation process of APM. We make an ablation study to verify their contributions to APM.

**[Experiment on other Multi-modal dataset (qAEJ)]**

We extend APM from the image-text dataset to the audio-text dataset to verify its effectiveness in different multimodal domains. To improve the reproducibility, we introduce the implementation details and release our code and logs in [anonymous GitHub](https://anonymous.4open.science/r/ICLR-735-MMDD).

&nbsp;

We believe we have addressed most of the reviewers' concerns. Thank you for considering this context together with the original reviews.

---

### Meta-Review · Area_Chair_AadC · 2026-01-09

**Summary:**

The paper introduces Analytic Parameter Matching (APM) to improve the efficiency of multi-modal dataset distillation by replacing trajectory matching with analytic alignment. While reviewers acknowledged the efficiency gains and the novel theoretical link to matrix whitening, the decision to reject is informed by persistent concerns regarding the method's strict reliance on linear projectors and the theoretical justification for using a simplified regression objective as a valid proxy for InfoNCE loss.

**Reviewer Concerns:**

The authors successfully addressed concerns regarding the lack of diverse downstream tasks (by adding zero-shot classification and audio-text experiments) and missing comparisons with generative methods (EDGE). However, outstanding concerns remain regarding the rigorousness of the theoretical derivation (simplifying InfoNCE to least-squares without considering temperature/negatives), the limited applicability to non-linear architectures, and the marginal performance gains over strong baselines like RepBlend.

**Reviewer Scores:**

Reviewer teJU would likely raise their score (from 4 to 6) as they explicitly mentioned improved presentation and clearer baselines. However, Reviewers P2Ji, q6bY, and qAEJ would likely maintain their borderline scores (hovering around 5 or weak 6), as the fundamental concerns about the method's soundness and generalization capability were not fully resolved by the rebuttal.

---

### Decision · Program_Chairs · 2026-01-26

Reject